# ECHO: Where Multilingual Sentence Embeddings Speak the Same Language

## Abstract

Cross-lingual sentence encoders create unified embedding representations of sentences across languages. However, achieving both strong downstream performance and cross-lingual alignment remains a fundamental challenge. Early models relied on contrastive learning, yet were unable to leverage hard negatives to unlock the full benefits of the contrastive paradigm. These contrastive approaches were surpassed by non-contrastive approaches leveraging token-level decoders. This is in contrast with recent generic embedding models that achieve strong results by combining contrastive objectives, large language models (LLMs) initialization, and hard negatives usage. We introduce ECHO, a novel cross-lingual sentence encoder that bridges this gap by integrating pretrained LLMs in an Encoder-Decoder architecture with contrastive training and hard negatives. Our bottleneck Encoder-Decoder design forces the model to capture essential semantic information in a shared vector space while preserving fine-grained nuances. ECHO achieves half the error rate of the previous state-of-the-art encoders in cross-lingual similarity search across 200 languages, while showcasing unprecedented cross-lingual transfer on downstream tasks.

## 1 Introduction

The development of multilingual models has long been a central focus in the field of Natural Language Processing, spanning applications from traditional Machine Translation (NLLB Team et al., 2022) to the recent surge in multilingual large language models (Workshop et al., 2022; Üstün et al., 2024; Team et al., 2025). A persistent challenge in this domain is the scarcity of training data for many languages. This has motivated research into cross-lingual representation learning (Devlin et al., 2019; Conneau et al., 2019; Janeiro et al., 2025a; Alastruey et al., 2025) that can generalize across languages and transfer the performance of resource-rich languages into lower resourced ones.

Among cross-lingual representations, cross-lingual sentence embeddings enable a vast array of applications, that would otherwise not be possible. From expanding multilingual coverage of language modeling, even while training on monolingual data, as shown in the LCM (LCM team et al., 2024), to large-scale cross-lingual similarity search for mining (Schwenk et al., 2021) that led to significant improvements in machine translation systems (NLLB Team et al., 2022). Aligned multilingual sentence embeddings demonstrate strong cross-lingual properties. They have recently been applied to a wider range of multilingual tasks, including classification (Costa-jussà et al., 2024) and translation quality estimation (Chen et al., 2023a; Dale & Costa-jussà, 2024). In general, since their representations are aligned across languages, they unlock multilingual zero-shot downstream performance for tasks without the need of data in all languages.

Early cross-lingual sentence encoders relied on contrastive signals (Feng et al., 2022; Yang et al., 2019) but failed to effectively leverage hard negatives. Recent alternatives such as SONAR (Duquenne et al., 2023) and MEXMA (Janeiro et al., 2025b), outperformed them by using translation reconstruction on top

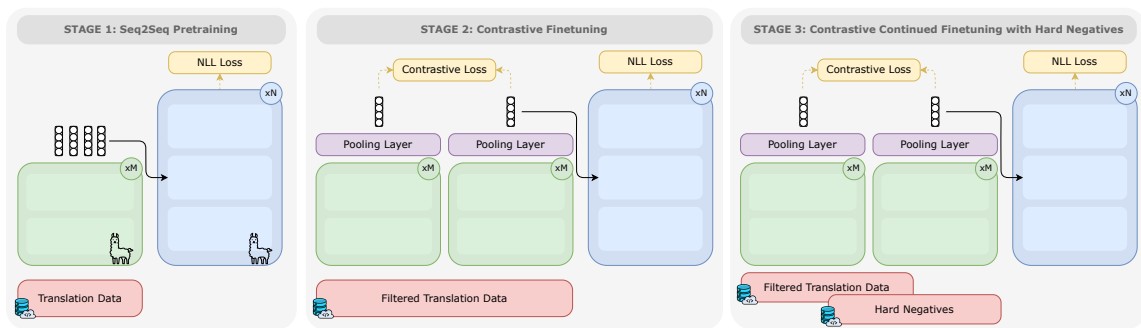

Figure 1: The ECHO method, divided into its 3 training stages. Stage 1 is Seq2Seq training with the translation objective. Stage 2 is contrastive alignment with translation. Stage 3 is contrastive with hard negatives.

of multilingual encoders. However, this approach diverges from the design principles of modern general-purpose embeddings, which typically combine contrastive losses with hard negatives (Wang et al., 2024b). Meanwhile, the success of Large Language Models (LLMs) has motivated a new paradigm of adapting them as encoders (BehnamGhader et al., 2024; Wang et al., 2024a; Zhang et al., 2025b) to take advantage of their extensive pre-training knowledge. While these approaches achieve impressive monolingual performance, they largely overlook cross-lingual transfer and alignment objectives. We present a novel training recipe that, for the first time, combines a translation loss from a decoder and a contrastive signal with hard negatives, to learn a language-agnostic sentence embedding space. Through a comprehensive analysis of learning objectives, we demonstrate the critical importance of each of these components alongside LLM initialization.

We present ECHO, a new state-of-the-art cross-lingual sentence embedding model that bridges the gap between strong performance and optimal cross-lingual alignment, along with a comprehensive analysis of the key components, including model architecture, data, and training objectives, that contribute to optimal cross-lingual properties in sentence embedding spaces.

Our main contributions are as follows:

- We adapt an English-centric LLM as both a largely multilingual encoder with bidirectional self-attention and a decoder for sequence-to-sequence modeling within a framework of sentence embedding learning.

- We couple a translation objective with a contrastive objective for alignment in a bottleneck encoder-decoder framework, where the encoder compresses multilingual input into a shared representation space.

- We enhance contrastive learning via online negatives removal, margin regularization, and a novel split softmax approach that separately optimizes hard negatives and in-batch negatives.

- We present ECHO, a new state-of-the-art embedding model covering 200 languages that achieves superior performance in multilingual alignment and cross-lingual transfer, as demonstrated through comprehensive evaluation on downstream tasks.

- We conduct extensive ablation studies to analyze the contribution of each component.

## 2 RELATED WORK

The field of multilingual sentence embeddings has grown rapidly, driven by benchmarks like MTEB (Muennighoff et al., 2023), xsim/xsim++ (Artetxe & Schwenk, 2019; Chen et al., 2023b), and MIRACL (Zhang et al., 2023).

**MULTILINGUAL ALIGNMENT** Multilingual aligned embedding models map vector representations across languages into shared spaces. Training on translation data typically enables semantic alignment via contrastive objectives using encoders only (Feng et al., 2022; Yang et al., 2019; Miao et al., 2024) or non-contrastive objectives with decoder signals (Janeiro et al., 2025b; Duquenne et al., 2023). In ECHO, we combine both decoder and contrastive losses.

**CONTRASTIVE LEARNING** While contrastive learning dominates sentence embedding training (Gao et al., 2021), hard negatives remain underexplored in multilingual alignment, with LaBSE (Feng et al., 2022) reporting negative results. General purpose models (Wang et al., 2024b; Sturua et al., 2024) have successfully used mined and synthetic negatives. With ECHO we unlock contrastive objectives with synthetic hard negatives for better multilingual alignment.

**CODE AND MATH** Recent general purpose models (Wang et al., 2024b; Nussbaum & Duderstadt, 2025) and code-specific embeddings (Zhang et al., 2024; Suresh et al., 2025; Liu et al., 2024) incorporate code and math data. Most code embedding systems use docstring-implementation pairs (Husain et al., 2019; Zhang et al., 2024; Suresh et al., 2025), focusing on function-level rather than sentence-level representations.

## 3 DATA PROCESSING

**TRANSLATION DATA** NLLB (NLLB Team et al., 2022) has become the standard source of paired translation data for learning multilingual sentence embeddings (Duquenne et al., 2023; Wang et al., 2024b; Janeiro et al., 2025b), offering coverage of up to 200 languages and more than 40 billion paired examples. We use both human-labeled, mined and back-translated data from NLLB to train ECHO. To further broaden our coverage for lower resourced languages and word-level representations, we incorporate word-level dictionary data from PanLex (Kamholz et al., 2014) and add more than 3K language pairs directions of word translations. Our final natural language translation data is constructed by sampling from the original NLLB data, supplemented with dictionary-based pairs. Statistics for each split are presented in Appendix Table 7. As this data is inherently paired, it can be directly leveraged in our experimental setup. However, it does not naturally include per-sample negatives, a limitation we address through synthetic data generation in subsequent stages.

**CODE AND MATH DATA** Although our primary focus is on sentence-level, modality-agnostic representations, we treat code and mathematical expressions as semantic units that can be mapped into this shared embedding space. In this framework, programming languages like JavaScript or Go are considered alongside natural languages such as Catalan or Portuguese. To construct translation data including both programming and natural languages, we develop a comprehensive pipeline that addresses the limitations of traditional docstring-based approaches. We focus on sentence-level code snippets and mathematical expressions whose semantics can be described in a single natural language sentence. Our approach involves: (1) syntax-aware segmentation of code from 7 programming languages using Abstract Syntax Trees, (2) extraction of LaTeX mathematical expressions from scientific corpora, (3) generation of natural language descriptions using LLaMA3.3 70B Instruct, and (4) creation of multilingual versions through back-translation. Quality is ensured through consistency filtering of the synthetic data. For complete technical details, implementation procedures, and filtering methods, please refer to Appendix A.1.

**DATA FILTERING**   As detailed in Section 4, we train ECHO in multiple stages. While the first stage uses a large amount of data, later contrastive and hard-negative training stages require fewer steps and less data. We therefore reduce data volume using quality estimation signals. For natural language data, we apply BLASER 2.0 (Dale & Costa-jussà, 2024) filtering, while for code and math data, we subsample. We focus on X-to-English directions as they are the most populated and facilitate hard negative generation. For each language direction, we select the top 1 million pairs from human-labeled NLLB data, supplementing with highest-scoring mined and backtranslated pairs when needed to reach the 1 million threshold. Data statistics are reported in Appendix Table 7.

**HARD NEGATIVES GENERATION**   We leverage both in-batch and hard negatives for contrastive training. Based on the intuition behind Chen et al. (2023b), the ideal hard-negative for a translation pair is an approximate paraphrase of the original translation but with a subtle or traditionally hard to encode semantic modifier. We synthetically generate these hard negatives using LLaMA3.3 70B Instruct. For more details see Appendix A.2.

## 4   MODEL

In this section, we describe our model and method for training the ECHO embedding space. The whole training procedure is depicted in Figure 1, and is comprised of three different parts.

### 4.1   ARCHITECTURE, INITIALIZATION AND TOKENIZER

We use a bottleneck encoder-decoder architecture based on the transformer architecture (Vaswani et al., 2017), following the SONAR approach (Duquenne et al., 2023). We repurpose the architecture from LLaMA3 (Grattafiori et al., 2024) for our transformer architecture and use an embedding representation of 1024 dimensions. Inspired by previous work (BehnamGhader et al., 2024; Zhang et al., 2025a), we initialize both the encoder weights and the decoder weights with LLaMA3 (Grattafiori et al., 2024). We replace the causal self-attention in the encoder by bi-directional self-attention (BehnamGhader et al., 2024). We add cross-attention blocks in the ECHO decoder to attend to encoder outputs. The cross-attention weight matrices are randomly initialized.

Initializing our model with LLaMA3 weights constrains us to use LLaMA3 tokenizer. To increase its multilingual coverage, we extended the LLaMA3 vocabulary from 128k to 256k tokens for better fertility across our 200 target languages. Details about the tokenizer vocabulary extension are given in Appendix D. We initialize the embeddings for the new tokens by tokenizing them with the original tokenizer and averaging resulting token embeddings to create the new token embedding (Gee et al., 2022; Moroni et al., 2025).

### 4.2   SEQ2SEQ PRETRAINING

Before learning the embedding space itself, we introduce a sequence-to-sequence (Seq2Seq) pretraining stage, to warm-up our encoder-decoder model on translation tasks (stage 1 in Figure 1). In this stage, encoder outputs are not pooled before being passed to the decoder. The model is trained with a translation objective - source sentences are fed to our model as encoder inputs, and we optimize cross-entropy loss between decoder outputs and target sentences. We jointly optimize all translation tasks – natural language, code and math – during this Seq2Seq pretraining stage, with more than 5 thousand translation directions.

To enable effective multilingual and multitask processing, we employ natural text prompting for both encoder and decoder inputs. Source sentences are prefixed with language identifiers using the format "[language name]:". Target sentences incorporate task specification, output language information, and data provenance (human-labeled translations, automatically extracted translations, or back-

translations), following NLLB Team et al. (2022). Specifically, we use the prompts such as `This is a possible translation in [language name]:` for translation tasks and `This is a possible natural language explanation in English:` for code and math explanation tasks. We provide the full list of prompts in Appendix Table 17.

### 4.3 CONTRASTIVE FINETUNING

Contrastive finetuning is stage 2 in Figure 1. In this stage, we initialize the encoder and decoder with the weights obtained in the Seq2Seq stage. Then, we align the pooled source and anchor representations outputted by the encoder. The anchor is the translation fed to the encoder. This is done through a Siamese network trained with a contrastive loss. Additionally, alongside the contrastive loss, we train our model on translation tasks with a cross-entropy loss between decoder outputs and target sentences, following SONAR (Duquenne et al., 2023). Contrarily to previous Seq2Seq training stage, the decoder only attends to the source pooled encoder representation instead of cross-attention on full encoder outputs. We perform *CLS* pooling with a new token prepended to each input to the encoder to create our fixed-size sentence representation.

Our contrastive objective, Equation (1), uses a modified InfoNCE loss (Chen et al., 2020). We add a margin to the similarity scores of source-positive pairs, following LaBSE (Feng et al., 2022), to make translations more distinct from non-translations in the resulting embedding space. The contrastive loss is defined as:

$$\mathcal{L}_{contrastive} = -\frac{1}{N}\sum_{i=1}^{N}\frac{e^{\phi(x_i,y_i)-m}}{e^{\phi(x_i,y_i)-m}+\sum_{n\in\mathcal{S}_i}e^{\phi(x_i,y_n)}} \tag{1}$$

where $\phi(x_i,y_i)$ denotes the scaled cosine similarity between a source sentence $x_i$ and a target sentence $y_i$, $\phi(x_i,y_i) = cos(x_i,y_i)*\tau$, with $\tau$ being a logit scaling hyperparameter, and $m$ is an additive margin hyperparameter applied to the source-positive pairs.

Negative examples are drawn from in-batch samples, but we filter them to ensure that no false negatives are used, following GISTEmbed (Solatorio, 2024). Specifically, the set of negatives $S_i$ for each source $x_i$ is defined as:

$$\mathcal{S}_i = \{\, j \in \{1,\ldots,N\} \mid \phi(\overline{x_i},\overline{y_j}) < r \cdot \phi(\overline{x_i},\overline{y_i})\,\} \tag{2}$$

where $r$ is the hyperparameter for the radius of negatives removal and $\overline{x_i}/\overline{y_j}$ are guide embeddings given by SONAR (Duquenne et al., 2023). This filtering step removes any negative whose similarity to the source exceeds that of the positive pair, ensuring that the model does not learn from negatives that are more similar to the source than the true translation.

The training loss is then the combination of the contrastive and the decoder loss:

$$\mathcal{L} = \alpha \cdot \mathcal{L}_{contrastive} + \beta \cdot \mathcal{L}_{translation} \tag{3}$$

where $\alpha$ and $\beta$ are hyper-parameters that control the weight of each loss term.

### 4.4 CONTRASTIVE CONTINUED-FINETUNING WITH HARD NEGATIVES

To further improve the model's ability to distinguish between close translations, we perform an additional contrastive step using hard negatives (stage 3 in Figure 1). The hard negative generation is described in Section 3. Initial experiments showed that, contrary to in-batch negatives, contrastive learning with non-zero additive margin was not effective with hard negatives. In order to simultaneously optimize contrastive learning involving hard and in-batch negatives, we introduce an additional separate contrastive loss to handle hard negatives. This enables us to weight the contribution of in-batch contrastive loss and hard-negative contrastive loss without margin. The resulting loss is then defined as:

$$\mathcal{L}_{contrastive\_hn} = (1-\gamma) \cdot \mathcal{L}_{contrastive} - \gamma \cdot \frac{1}{N}\sum_{i=1}^{N}\frac{e^{\phi(x_i,y_i)}}{e^{\phi(x_i,y_i)}+\sum_{h_j\in\mathcal{S}_i^{\text{HN}}}e^{\phi(x_i,h_j)}} \tag{4}$$

where $\mathcal{S}_i^{\text{HN}}$ is the list of hard negatives for source $x_i$, and $\gamma$ is the objective contribution weighting hyperparameter. Our overall loss is now $\mathcal{L} = \alpha \cdot \mathcal{L}_{contrastive\_hn} + \beta \cdot \mathcal{L}_{translation}$.

### 4.5 DECODER FINETUNING

SONAR is composed of an encoder and a decoder. The availability of the decoder, despite not being a pre-requisite for an embedding model, enables to efficiently decode sentence embeddings into natural text in several languages. This was proven useful in some new research directions like Language Modeling in sentence embedding spaces (LCM team et al., 2024) where predicted embeddings are decoded into text. ECHO also leverages a decoder during training, as explained in previous sections. To enhance the decoder performance for downstream use, we continue its learning on top of ECHO obtained after Section 4.4. We initialize both encoder and decoder weights from that training stage but freeze the encoder parameters. We then use the same loss and data setup as in Section 4.2.

### 4.6 EXPERIMENTAL CONFIGURATION

**SEQ2SEQ** We train our model for 100k steps in this stage, with 8192 tokens per GPU trained across 16 nodes of 8 GPUs each. The encoder and decoder are initialized from LLaMA3.2 1B size, trained with fsdp1 and mixed precision on fp16, with a maximum gradient norm of 1. We use the AdamW optimizer with betas 0.9 and 0.98. Our learning rate is set to 4e-4, with 2k warmup steps and Myle learning rate scheduler.

**CONTRASTIVE FINETUNING** Unless specified, the parameters are the same as the Seq2Seq configuration described above. For contrastive tuning we change the learning rate to 3e-4, max number of tokens per GPU to 6k, and set the contrastive loss weight, $\alpha$, to 0.05, with the translation loss weight, $\beta$, being 1. We define our radius for false negatives removal, $r$, to 0.5, our margin, $m$ to 0.3 and our scale $\tau$ to 100. Our model is trained for 10k steps.

**CONTRASTIVE CONTINUED-FINETUNING WITH HARD NEGATIVES** We take 5 hard negatives per source sentence, and change the max number of tokens to 1.2k (6k/5). The learning rate is changed to 1e-5, with 15k steps. $\gamma$, the weight between the in-batch and the hard negative objectives, is defined as 0.8.

**DECODER FINETUNING** We use same training setup as the Seq2Seq training stage except for learning rate which is set to 1e-3 and number of warmup steps which is lowered to 200.

## 5 RESULTS

In this section, we present results obtained with ECHO on cross-lingual similarity search, downstream classification and pair classification tasks, as well as cross-lingual transfer quantification.

### 5.1 MULTILINGUAL ALIGNMENT - BITEXT MINING

To evaluate cross-lingual alignment, we perform similarity search on FLORES translations (NLLB Team et al., 2022), comparing source sentence embeddings to candidate translation pools. We report error rates as xsim (mining non-English sentences against English translations) and xsim++ (Chen et al., 2023b), which adds English hard negatives.

Table 1 presents results for ECHO and competitive baselines on both commonly supported languages (Table 6) and all FLORES languages for fair comparison. ECHO achieves state-of-the-art performance, with significant improvements in xsim and xsim++ (7.15% absolute improvement over 200 languages), indicating

|  | common languages | | all languages | |
| --- | --- | --- | --- | --- |
| model | xsim ↓ | xsim++ ↓ | xsim ↓ | xsim++ ↓ |
| MEXMA | 0.08 | 7.80 | 15.91 | 35.78 |
| LaBSE | 2.39 | 23.35 | 18.61 | 48.69 |
| mE5$_{large}$ | 0.62 | 23.87 | 9.31 | 39.32 |
| SONAR | 0.17 | 9.88 | 1.37 | 15.27 |
| ECHO | **0.07** | **3.90** | **0.99** | **8.12** |

Table 1: xsim/xsim++ results for all models on FLORES devtest, as X-eng cross-lingual similarity search.

better semantic alignment and robustness to hard negatives through improved handling of lexical and semantic nuances. We report complete breakdowns of xsim/xsim++ evaluation across languages in Appendix F.

We further evaluate on GMMLU (Singh et al., 2024), MMLU translated to 42 languages, by pairing questions in any language to their English equivalent and XLCoST (Zhu et al., 2022), to our knowledge the only snippet-level Code2Code benchmark.

Table 2 shows ECHO outperforms all systems on GMMLU except MEXMA on common languages, but leads across all 42 languages. Notably, ECHO surpasses specialized code-embedding models like CodeSage Zhang et al. (2024) and CodeRankEmbed (Suresh et al., 2025) on XLCoST, excelling at code representation even for unseen programming languages like C#.

## 5.2 DOWNSTREAM TASKS

To assess the quality and generalization of our embeddings we evaluate them on several multilingual classification and pair classification benchmarks under MTEB (Muennighoff et al., 2023), see Table 8 for full list. Results are reported in Table 3.

**CLASSIFICATION** The reported metric for classification is accuracy. Under this setup, linear classifiers are trained on top of each model's embeddings on a held-out portion of the data, and evaluated on the rest. Each classifier is trained and evaluated per language in this section. Our reported numbers are first averaged over all languages in each benchmark and then over all benchmarks to create a single score. Table 3 shows how ECHO far outperforms all other models in classification tasks, highlighting the good content in each individual vector, and as we will explore in future sections, their interoperability across languages.

| Model | GMMLU (all) | GMMLU (common) | C | C++ | C# | Java | Javascript | PHP | Python | All |
| --- | --- | --- | --- | --- | --- | --- | --- | --- | --- | --- |
| MEXMA | 6.97 | **1.26** | 18.87 | 24.53 | 22.22 | 22.91 | 20.98 | 16.14 | 24.06 | 21.39 |
| LaBSE | 3.43 | 2.95 | 19.84 | 27.35 | 24.31 | 24.92 | 24.20 | 22.07 | 26.25 | 24.13 |
| SONAR | 3.18 | 2.96 | 22.03 | 29.39 | 28.34 | 29.40 | 26.01 | 22.23 | 30.82 | 26.89 |
| ECHO | **2.02** | 1.70 | **15.60** | **20.02** | **19.17** | **18.99** | **17.28** | **13.00** | **18.57** | **17.52** |
| mE5$_{large}$ | 5.31 | 3.27 | 16.36 | 22.42 | 20.48 | 20.39 | 18.45 | 13.53 | 20.14 | 18.82 |
| CodeSage-large-v2 | – | – | 19.41 | 23.02 | 21.17 | 21.47 | 18.19 | 15.50 | 20.42 | 19.89 |
| CodeRankEmbed | – | – | 16.71 | 21.48 | 19.85 | 20.47 | 17.36 | 13.40 | 19.67 | 18.42 |

Table 2: Results for GMMLU question mining (left) for all 42 languages and those covered by the baselines (common) and XLCOST (right). xsim (↓) reported for all models.

| Model | Average | Classification | Pair Classification |
|---|---|---|---|
| MEXMA | 65.895 | 68.690 | 63.100 |
| LaBSE | 65.205 | 65.770 | **64.640** |
| SONAR | 64.325 | 67.910 | 60.740 |
| ECHO | **67.720** | **72.200** | 63.240 |
| General-purpose models | | | |
| mE5$_{large}$ | 70.570 | 68.260 | 72.880 |

Table 3: Classification and Pair Classification results from sentence-level MTEB tasks.

**PAIR CLASSIFICATION** For Pair Classification we report on average precision based on the cosine similarity between pairs. In this case we see how ECHO still outperforms multilingual embedding models in its category, with the exception of LaBSE, while it lags behind the topline comparison of mE5$_{large}$ which was trained as a general-purpose embedding model. It is important to highlight that all our baselines along with ECHO are trained solely on parallel data, i.e. no task specific data is involved and the cosine distance between sentences reflects just that aspect.

## 5.3 CROSS LINGUAL TRANSFER

We evaluate alignment across languages in the lens of classification. Namely, we train a classifier to classify French sentences from the SIB200Classification task in MTEB and apply it, in a zero-shot fashion, to the other 199 languages in SIB. We report Cross-lingual transfer (CLT) ratio in Table 4, which corresponds to the ratio of classification accuracy for language L with classification accuracy on French. This table highlights the strong cross-lingual transfer with ECHO representations across 200 languages, exceeding 97% average CLT ratio over 200 languages, and over 99% over the most common 80 languages.

## 5.4 DECODING CAPABILITIES

Decoding sentence embeddings into natural text can help quantify the text compression ability of the embedding model across languages. The decoding results remain nonetheless dependent on the decoder training and capacity, in addition to the sentence embedding representations themselves. Moreover, models that predict sentence embeddings, like Large Concept Models (LCM team et al., 2024), rely on the ability of good text decoders to produce text in many languages. Therefore, we report translation results, as measured by spBLEU (Post, 2018) (with *flores200* tokenizer) and chrF++ (Popović, 2017), on FLORES devtest based

| model | SIB200 CLT ratio | |
|---|---|---|
| | all | common |
| LaBSE | 80.58% | 91.99% |
| MEXMA | 78.38% | 95.56% |
| mE5$_{large}$ | 84.76% | 95.47% |
| SONAR | 92.34% | 96.22% |
| ECHO | **97.15%** | **99.26%** |

Table 4: Cross-lingual transfer (CLT) on SIB200Classification: Models trained on French, evaluated zero-shot on 199 languages (all) and 80 baseline-supported languages (common), reporting average relative performance to French.

| | X to English | | English to X | |
|---|---|---|---|---|
| model | spBLEU | chrF++ | spBLEU | chrF++ |
| SONAR | 32.62 | 54.79 | 20.29 | 42.71 |
| ECHO | **33.27** | **55.02** | **21.17** | **43.63** |

Table 5: Average translation performance of SONAR and ECHO on FLORES devtest set for X to English and English to X directions, as measured by spBLEU and chrF++ metrics. Source sentences are embedded into the sentence embedding space before being decoded into the target language with their decoder.

on ECHO model and compare them with SONAR translation results (SONAR being the only multilingual sentence embedding space coming with a decoder) in Table 5. ECHO shows significantly better translation performance compared to SONAR on this decoding task.

## 6 ANALYSIS AND ABLATIONS

ECHO's design choices are validated through ablation studies showing significant improvements at each stage. Adding the Decoder loss to the contrastive learning stage in subsection 4.3 reduces xsim++ error by $45\%$ (from 16.23 to 8.95), demonstrating that token-level language modeling signals capture semantic nuance beyond surface features. Replacing MSE with contrastive loss improves xsim++ by $29\%$ (from 12.54 to 8.95), as contrastive learning creates more structured embedding spaces by explicitly separating negatives, a key difference with previous approaches such as SONAR. Using separate losses for in-batch and hard negatives (split softmax in subsection 4.4) improves xsim by $19\%$ compared to a single softmax approach, preventing convergence issues while better balancing negative types. Finally, initializing contrastive learning from our Seq2Seq-adapted model rather than directly from LLaMA reduces xsim error by $36\%$ and xsim++ by $25\%$, showing the multilingual adaptation stage provides a better foundation. Additionally, we show that we can have smaller models with minimal performance degradation.

For additional ablations and complete experimental details, see Appendix C.

## 7 CONCLUSION

In this work, we introduced a state-of-the-art cross-lingual sentence encoder, ECHO. We got closer to the stated goal of creating a language-agnostic space in which sentences with same semantic meaning share vector representations, regardless of the language. Compared to previous efforts, ECHO shines in its multilingual alignment where the error rates are halved. This enables downstream tasks, especially for lower resource languages where all previous models lacked behind. At the same time ECHO outperforms all comparable baselines in downstream evaluations, closing the gap with general-purpose embedding models such as mE5$_{\text{large}}$ that fail in their cross-lingual transfer and alignment. Moreover, task-specific modules trained on the rich space ECHO provides, require only training in a single language and seamlessly transfer to the others. We are excited about the new uses such an embedding space will create.

Our works proposes a new training paradigm for text embedding **representation learning**, where decoders should be coupled with a contrastive loss objective for improved performance. With an extensive set of ablations, we pave the way of new training recipes on top of Large Language Models to transform them into embedding Encoders. Most current efforts focused on expanding attention, pooling a representation for the text, and training it on a contrastive signal; an effective yet unexciting recipe. The addition of a Decoder is key to capture fine-grained features within the embeddings. While we focused on training sentence-level

language-agnostic embeddings using translation data, we believe future work should exploit our framework for general-purpose embeddings.

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

## A    DATA PROCESSING

### A.1    CODE AND MATH TRANSLATION DATA GENERATION

#### A.1.1    CODE SNIPPET SEGMENTATION

To construct sentence-level code snippets suitable for embeddings, it is essential to define what constitutes a "sentence" in the context of programming languages. Unlike natural language, where sentences are typically delimited by punctuation, code structure is governed by syntax and semantics, making naive approaches, such as splitting at line breaks, insufficient and potentially misaligned with real-world coding practices.

To address this, we adopt a syntax-aware segmentation strategy similar to Gong et al. (2025), leveraging Abstract Syntax Trees (ASTs) to identify meaningful breakpoints within code. This approach allows us to segment code in a way that respects its logical and syntactic boundaries, rather than relying on superficial heuristics. For our experiments, we use code from seven programming languages (Python, Java, JavaScript, Go, C, C++, and Ruby) sourced from publicly licensed GitHub repositories.

Our segmentation process begins by parsing source code into an AST using the Tree-sitter library[1]. We then traverse the tree in reverse Breadth-First Search (BFS) order, starting from the leaf nodes and progressing bottom-up. For each node, if it is a leaf with non-empty text and has not yet been visited, we initiate a snippet. We classify the snippet as either "code" or "text" based on the node type (e.g., comments and strings are labeled as "text").

To form coherent and contextually meaningful snippets, we recursively expand each snippet upward by merging the parent statement or declaration and its unvisited children, provided that the combined size does not exceed a maximum threshold of 100 non-whitespace characters. This ensures that each snippet remains concise and suitable for sentence-level representation. The process continues until all nodes have been visited, resulting in a comprehensive set of segmented code snippets.

The full segmentation procedure is detailed in Algorithm 1, which outlines the AST traversal, snippet formation, classification, and postprocessing steps. This method enables us to extract sentence-level code snippets that are both syntactically coherent and semantically meaningful, facilitating their integration into our modality-agnostic embedding space.

#### A.1.2    MATH EXPRESSIONS GATHERING

To build a high-quality dataset of mathematical expressions, we extract LaTeX math content from large-scale scientific corpora such as FineMath Allal et al. (2025) and arXiv. Our extraction process is designed to capture both inline and display math, reflecting the diversity of mathematical notation found in scientific writing. The expressions used can be found in Table 2. We use a comprehensive set of regular expressions to identify a wide range of LaTeX math environments. To ensure the quality and relevance of the extracted expressions, we apply the following filters:

- Expressions between 20 and 150 characters.

- Expressions where more than 90% of non-whitespace characters are alphabetic are discarded, except for in-line math.

The resulting dataset consists of unique LaTeX mathematical expressions, both in isolation and within their natural language in-line context, providing a rich resource for training and evaluating modality-agnostic sentence-level embeddings.

---

[1]https://github.com/tree-sitter/tree-sitter

---

**Algorithm 1** Code Segmentation via Abstract Syntax Tree Traversal

---

**Require:** Source code, language parser, segmentation parameters (max size, depth, etc.)
**Ensure:** List of code segments (character ranges, types)
1: **Parse** source code → syntax tree (using `https://github.com/tree-sitter/tree-sitter`)
2: **Initialize** empty list of snippets, visited node set
3: **for** each tree level (BFS order), processed in reverse order (bottom-up) **do**
4:     **for** each node at this level **do**
5:         **if** node is a leaf, has non-empty text, and is not visited **then**
6:             snippet ← {node}
7:             **Classify** snippet type:
8:             **if** node type is comment or string **then**
9:                 snippet_type ← "text"
10:            **else**
11:                snippet_type ← "code"
12:            **end if**
13:            **while** expansion upward is allowed (size and depth constraints not exceeded) **do**
14:                **if** parent node is a statement/declaration and adding it (and its children) keeps snippet size within allowed maximum **then**
15:                    snippet ← snippet ∪ parent node ∪ eligible siblings
16:                    **Update** snippet_type if parent changes classification
17:                **else**
18:                    **break**
19:                **end if**
20:            **end while**
21:            Mark included nodes as visited
22:            Add (snippet, snippet_type) to output list
23:        **end if**
24:     **end for**
25: **end for**
26: **Postprocess:**
27: Merge adjacent snippets if their combined size is below the threshold and they are contiguous
28: Adjust segment boundaries to snap to whitespace or newlines as configured
29: **for** each snippet **do**
30:     Compute snippet's character range in source code
31: **end for**
32: **return** list of snippet ranges, snippet types

---

| Pattern | Description |
|---|---|
| `$(.*?)$` | Inline math (e.g., `$a^2 + b^2 = c^2$`) |
| `$$(.*?)$$` | Display math with double dollar signs |
| `\\[(.*?)\\]` | Display math with `\[ ... \]` |
| `\\begin{equation}(.*?)\\end{equation}` | Equation environment |
| `\\begin{align}(.*?)\\end{align}` | Align environment |
| `\\begin{align*}(.*?)\\end{align*}` | Align* environment |
| `\\begin{multline}(.*?)\\end{multline}` | Multline environment |
| `\\begin{multline*}(.*?)\\end{multline*}` | Multline* environment |
| `\\begin{gather}(.*?)\\end{gather}` | Gather environment |
| `\\begin{gather*}(.*?)\\end{gather*}` | Gather* environment |
| `\\begin{eqnarray}(.*?)\\end{eqnarray}` | Eqnarray environment |
| `\\begin{eqnarray*}(.*?)\\end{eqnarray*}` | Eqnarray* environment |
| `(?<=[.!?])\s+` | Sentence splitting after ., !, or ? |
| `(?<!\$)\$[^$]+\$(?!\$)` | Short inline LaTeX expressions |

Figure 2: Summary of regular expressions used for extracting LaTeX math expressions and splitting sentences.

### A.1.3 NATURAL LANGUAGE DESCRIPTION GENERATION

We leverage Llama-3.3-70B-Instruct to generate natural language descriptions for both code snippets and mathematical expressions. The model's extensive training on code and mathematical content enables effective paraphrasing of technical content into clear English descriptions. Importantly, this task involves paraphrasing existing content rather than generating new information. The prompts used for this generation process are shown in Figure 3.

### A.1.4 MULTILINGUAL BACK-TRANSLATION

To expand coverage of mixed-modality data, particularly sentences with inline expressions, we generate back-translations using Llama-3.3-70B-Instruct. We translate English descriptions and mixed-mode sentences into seven target languages: French, German, Hindi, Italian, Portuguese, Spanish, and Thai. This process creates a comprehensive multilingual dataset that enhances the diversity and utility of our training data while maintaining semantic consistency across languages.

### A.1.5 CONSISTENCY FILTERING

To validate the quality of our synthetic code-to-text pairs, we implement a consistency check using the CodeRankEmbed embedding model (Suresh et al., 2025). This process verifies that generated English descriptions accurately capture the semantics of their corresponding code snippets.

For each generated English description, we use it as a query to retrieve the most semantically similar code snippet from a pool of 100,000 candidates within the same programming language. If our synthetic data generation is effective, the English description should retrieve its original corresponding code snippet as the top match.

We find that in 99% of cases, the English description successfully retrieves its original code snippet as the top-1 match. This high retrieval accuracy indicates strong semantic alignment between code snippets and their generated natural language descriptions, demonstrating the reliability and fidelity of our synthetic data generation approach.

**Math Text Translation**

**System prompt:**
```
You are a helpful translation assistant.  You respond only with the
translation, without additional comments, context, or explanation.
```

**User prompt:**
```
Translate the following text from English into {target_lang}.  Don't
produce any other output outside of the translation.
{example}
```

**Code Snippet Translation**

```
Translate the following {programming_language} snippet to a single
sentence, ensuring that all elements and operations in the code are
included.  The sentence should convey the semantic meaning of the
code, effectively translating it into a clear and concise lexical
explanation without making any assumptions or inferences beyond what
is explicitly stated in the code.  Describe only and exactly its
explicit elements and operations, without any additional context or
explanation.  Use a single, direct sentence that includes all elements
and operations in the code, avoiding introductory words or additional
context.  Please provide only the sentence and nothing else:
{example}
```

**Math Formula Translation**

```
Describe the following mathematical text in a single sentence.  The
sentence should convey the semantic meaning of the mathematical
notation, effectively translating the mathematical notation into
a clear and concise lexical explanation.  Please provide only the
sentence and nothing else.
{example}
```

Figure 3: Three prompt templates for translation, code snippet semantic description, and mathematical notation explanation.

## A.2 HARD NEGATIVES GENERATION

For hard negatives generation we follow two strategies:

**Natural Language**  For natural language (i.e. no code or math) translation, we generate hard negatives using Llama 3.3 70B Instruct. We follow an approach inspired by xsim++ negatives Chen et al. (2023b), where they crafted hard-to-distinguish negative examples for translation pairs. We use the prompt described in Figure 4 and generate up to 5 hard negatives per sample.

**Code and math**   Here we follow a more straightforward approach and mine hard negatives using the ECHO checkpoint trained in Section 4.3 before hard negatives are introduced. We mine the top 5 negatives over a pool of 200k candidates for each sample. An example is provided in Figure 5.

---

**Hard Negatives Generation**

**You are a text transformation specialist. Generate ONLY valid xsim++ transformations using these EXCLUSIVE methods: 1. CAUSALITY ALTERATION:**

- Add/remove negations (*"did not", "was not"*)
- Replace adjectives with antonyms (*"good"* → *"bad"*)
- Change modal verbs (*"may"* → *"will"*)

**2. ENTITY REPLACEMENT:**

- Swap proper nouns (people, locations, organizations)
- Replace pronouns (*he* → *she, they* → *we*)

**3. NUMBER ALTERATION:**

- Change quantities (*5* → *12*)
- Modify dates/times (*2023* → *2019*)
- Alter percentages (*15%* → *22%*)

**Follow these patterns from training examples:**
```
{few-shot examples}
```
**Now transform THIS SPECIFIC INPUT SENTENCE using the above patterns. Output ONLY a Python list of 1-5 modified sentences in this exact format:**

```
[
"Transformed sentence 1",
"Transformed sentence 2",
...
]
```

**Key requirements:**

1. Create 1-5 unique modified sentences
2. Maximize difference from original text
3. Mix transformation types where possible
4. Maintain grammatical correctness
5. Do NOT generate paraphrases, or synonyms
6. NEVER output empty strings
7. Output ONLY a Python list of strings
8. No explanations, headers, or additional text

**Input sentence to transform:** `{example}`

---

Figure 4: Prompt for generating xsim++ transformations with clear instructions and structure.

## A.3   LANGUAGES BREAKDOWN

Table 6 lists all the languages supported by ECHO and common for all other models.

---

**Code/Math Hard Negatives Example**

```
Python: temp = pd.read_csv(item, header = None, dtype = float)
```

English: The pandas library is used to read a csv file specified by the item variable into a temporary variable named temp with the header set to None and the data type set to float.
Hard Negatives:

1. The code reads data into a variable named df using the pandas function `read_csv`.

2. The variable `temp_mean` is assigned the result of pandas' `read_csv` function applied to the string conversion of config's attribute.

3. The pandas library is used to read a csv file named "data.csv" located in the "/data" directory into a variable named df using the `read_csv` function.

4. The pandas library, referred to as pd, reads a comma-separated values file named `'data/time_series_19-covid-Confirmed.csv'` into a variable named df using the `read_csv` function.

5. The pandas library, referred to as pd, reads a comma-separated values file named `'tmdb-movies.csv'` into a dataframe variable named df using the `read_csv` function.

---

Figure 5: Example of code/math hard negatives generation.

### A.4 DATA STATISTICS

We present the statistics of our training data for both pre-training and fine-tuning stages in Table 7.

## B DOWNSTREAM TASKS

In Table 8, we have all MTEB tasks we use to evaluate the several models considered.

## C ABLATIONS AND ANALYSIS

ECHO includes several novel design choices supported by strong downstream performance. In this section we provide ablations for such choices in an incremental fashion, that lead to our final model reported in Section 5. All ablations experiments are trained for 5k steps only.

### C.1 TRAINING OBJECTIVES

ECHO follows a multi-stage training strategy described in Section 4. Some of these steps such as decoding loss for sentence embedding learning (Duquenne et al., 2023), LLM re-purposing as an Encoder-Decoder (Zhang et al., 2025a), and contrastive learning have been explored in isolation in prior work, but ECHO is the first system to train an embedding model with such training strategies in a unified framework. Here, we analyze the contribution of each component to the final performance.

As shown in Table 9a, each training stage yields significant improvements. After Seq2Seq pre-training, the representations are not yet optimized for sentence-level tasks, and mean-pooling over all tokens results in suboptimal performance. Nevertheless, we will later show the impact of this step as a foundation for subsequent contrastive training.

| Languages | | | | |
|---|---|---|---|---|
| ace_Arab | ace_Latn | **acm_Arab** | acq_Arab | **aeb_Arab** |
| **afr_Latn** | ajp_Arab | aka_Latn | als_Latn | **amh_Ethi** |
| apc_Arab | arb_Arab | ars_Arab | **ary_Arab** | **arz_Arab** |
| **asm_Beng** | ast_Latn | awa_Deva | ayr_Latn | **azb_Arab** |
| **azj_Latn** | bak_Cyrl | bam_Latn | ban_Latn | **bel_Cyrl** |
| bem_Latn | **ben_Beng** | bho_Deva | bjn_Arab | bjn_Latn |
| bod_Tibt | **bos_Latn** | bug_Latn | **bul_Cyrl** | **cat_Latn** |
| ceb_Latn | **ces_Latn** | cjk_Latn | **ckb_Arab** | crh_Latn |
| **cym_Latn** | **dan_Latn** | **deu_Latn** | dik_Latn | dyu_Latn |
| dzo_Tibt | **ell_Grek** | **eng_Latn** | **epo_Latn** | **est_Latn** |
| **eus_Latn** | ewe_Latn | fao_Latn | fij_Latn | **fin_Latn** |
| fon_Latn | **fra_Latn** | fur_Latn | fuv_Latn | gaz_Latn |
| **gla_Latn** | **gle_Latn** | **glg_Latn** | grn_Latn | **guj_Gujr** |
| hat_Latn | **hau_Latn** | **heb_Hebr** | **hin_Deva** | hne_Deva |
| **hrv_Latn** | **hun_Latn** | **hye_Armn** | ibo_Latn | ilo_Latn |
| **ind_Latn** | **isl_Latn** | **ita_Latn** | **jav_Latn** | **jpn_Jpan** |
| kab_Latn | kac_Latn | kam_Latn | **kan_Knda** | kas_Arab |
| kas_Deva | **kat_Geor** | **kaz_Cyrl** | kbp_Latn | kea_Latn |
| khk_Cyrl | **khm_Khmr** | kik_Latn | kin_Latn | **kir_Cyrl** |
| kmb_Latn | kmr_Latn | knc_Arab | knc_Latn | kon_Latn |
| **kor_Hang** | **lao_Laoo** | lij_Latn | lim_Latn | lin_Latn |
| lit_Latn | lmo_Latn | ltg_Latn | ltz_Latn | lua_Latn |
| lug_Latn | luo_Latn | lus_Latn | lvs_Latn | mag_Deva |
| mai_Deva | **mal_Mlym** | **mar_Deva** | min_Latn | **mkd_Cyrl** |
| mlt_Latn | mni_Beng | mos_Latn | mri_Latn | **mya_Mymr** |
| **nld_Latn** | **nno_Latn** | **nob_Latn** | **npi_Deva** | nso_Latn |
| nus_Latn | nya_Latn | oci_Latn | ory_Orya | pag_Latn |
| pan_Guru | pap_Latn | pbt_Arab | pes_Arab | plt_Latn |
| **pol_Latn** | **por_Latn** | prs_Arab | quy_Latn | **ron_Latn** |
| run_Latn | **rus_Cyrl** | sag_Latn | **san_Deva** | sat_Beng |
| scn_Latn | shn_Mymr | **sin_Sinh** | **slk_Latn** | **slv_Latn** |
| smo_Latn | sna_Latn | **snd_Arab** | **som_Latn** | sot_Latn |
| **spa_Latn** | srd_Latn | **srp_Cyrl** | ssw_Latn | **sun_Latn** |
| **swe_Latn** | **swh_Latn** | szl_Latn | **tam_Taml** | taq_Latn |
| taq_Tfng | tat_Cyrl | **tel_Telu** | tgk_Cyrl | tgl_Latn |
| **tha_Thai** | tir_Ethi | tpi_Latn | tsn_Latn | tso_Latn |
| tuk_Latn | tum_Latn | **tur_Latn** | twi_Latn | tzm_Tfng |
| **uig_Arab** | **ukr_Cyrl** | umb_Latn | **urd_Arab** | uzn_Latn |
| vec_Latn | **vie_Latn** | war_Latn | wol_Latn | **xho_Latn** |
| ydd_Hebr | yor_Latn | yue_Hant | zho_Hans | **zho_Hant** |
| zsm_Latn | zul_Latn | | | |

Table 6: Complete list of languages covered by our model. Languages shown in **bold** are supported by all models in our comparison. Our model covers 202 languages total, with 81 languages supported across all compared models.

A key distinction between ECHO and other embedding models built on modern LLMs is the inclusion of a Decoder component. While contrastive learning alone achieves a modest xsim score, it falls short on

| | Seq2Seq | | | | Contrastive | | | |
|---|---|---|---|---|---|---|---|---|
| Dataset | pairs | dirs | source | target | pairs | dirs | source | target |
| BT Math | 13.0M | 14 | 8 | 8 | – | – | – | – |
| Dictionary | 18.9M | 3.3K | 110 | 110 | – | – | – | – |
| Code/Math → Eng | 1.02B | 9 | 9 | 1 | 9.0M | 9 | 9 | 1 |
| Eng → Code | 941M | 8 | 1 | 8 | – | – | – | – |
| Eng → Math | 8.0M | 1 | 1 | 1 | – | – | – | – |
| NLLB Mined | 1.21B | 1.6K | 187 | 187 | 24.3M | 140 | 140 | 1 |
| NLLB mmt_bt | 901M | 258 | 132 | 128 | 107M | 119 | 119 | 1 |
| NLLB smt_bt | 215M | 76 | 39 | 39 | 3.2M | 37 | 37 | 1 |
| NLLB Primary | 398M | 1.3K | 202 | 202 | 51.6M | 196 | 196 | 1 |

Table 7: Training Data Statistics for Seq2Seq and Contrastive Learning Approaches. Each dataset shows: *pairs* (number of translation pairs), *dirs* (number of translation directions, i.e., language X to Y), *source* (number of source languages), and *target* (number of target languages).

| task | dataset |
|---|---|
| Classification | MassiveIntentClassification (FitzGerald et al., 2022) |
| | MassiveScenarioClassification (FitzGerald et al., 2022) |
| | MTOPDomainClassification (Li et al., 2021) |
| | MTOPIntentClassification (Li et al., 2021) |
| | AmazonCounterfactualClassification (O'Neill et al., 2021) |
| | SIB200Classification (Adelani et al., 2024) |
| Pair Classification | XNLI (Conneau et al., 2018) |
| | XNLIV2 (Upadhyay & Upadhya, 2023) |

Table 8: List of MTEB tasks we use to evaluate the models.

| Model | xsim | xsim++ |
|---|---|---|
| LLaMA initialization | 94.57 | 99.89 |
| Seq2Seq pre-training | 7.74 | 51.55 |
| Contrastive Loss | 0.71 | 16.23 |
| + Decoder Loss | 0.65 | 8.95 |
| + Hard negatives | 0.76 | 7.06 |

(a) Full method ablation.

| Model | xsim | xsim++ |
|---|---|---|
| Decoder + MSE losses | 0.92 | 12.54 |
| Decoder + Contrastive losses | 0.65 | 8.95 |

(b) Cross-lingual alignment objectives ablation.

Table 9: **Training objectives ablations**: Ablations on training objectives to learn a massively multilingual sentence embedding space on the cross-lingual similarity search task of FLORES200 dev set, as measured by xsim and xsim++.

xsim++. The addition of the cross-entropy loss from the Decoder, with its token-level language modeling signal, delivers the largest gains, highlighting its role in capturing semantic nuance beyond surface-level features. The introduction of hard negatives further reduces xsim++ scores.

|  | | xsim | xsim++ |
|---|---|---|---|
| **Margin** | 0 | 0.74 | 9.49 |
|  | 0.3 | 0.65 | 8.95 |
|  | 0.5 | 0.72 | 9.45 |
| **Logit scale** | 1 | 1.88 | 11.90 |
|  | 100 | 0.65 | 8.95 |
|  | 150 | 0.66 | 9.07 |
| **Gathering negatives** | no | 0.74 | 9.44 |
|  | yes | 0.65 | 8.95 |
| **False negative removal** | no | 0.69 | 9.73 |
|  | yes | 0.65 | 8.95 |

(a) Contrastive Learning hyper-parameters ablations

|  | xsim | xsim++ |
|---|---|---|
| *In-batch negatives only* | | |
| One softmax | 0.65 | 8.95 |
| *In-batch + hard negatives* | | |
| One softmax | 0.94 | 7.00 |
| Split softmax | 0.76 | 7.06 |

(b) Ablation on the use of hard negatives in contrastive learning.

Table 10: **Contrastive Learning ablations:** Effect of hyper-parameters and modeling options in Contrastive Learning on cross-lingual similarity search on FLORES200 dev set.

SONAR (Duquenne et al., 2023) successfully leveraged a Decoder to build sentence representations. However, their approach combined a Mean Squared Error (MSE) objective between source and target embeddings with the translation objective. In Table 9b, we show that replacing the MSE objective with a contrastive loss, as described in Section 4.3, leads to a substantial improvement. This result suggests that the contrastive signal encourages a more structured embedding space by explicitly pushing apart negatives, which benefits xsim++ and, as we discuss later, helps prevent embedding space collapse.

## C.2    CONTRASTIVE SIGNALS

Training embedding models with Contrastive Learning requires careful choices of hyper-parameters. We analyze the effect of these options on the cross-lingual similarity search results in Table 10.

The additive margin in the softmax improves separation between positive translations and negatives. A value of $m = 0.3$ was empirically found as best for this hyper-parameter, boosting performance compared to models trained without margin. We also explore the logit scale on cosine similarity, $\tau$, and find 100 to be the best and crucial for proper contrastive learning.

The choices of negative examples is also key. By default we use all other sentences from the batch as negatives, commonly referred to as in-batch negatives. We analyze the effect of different choices of negative examples in Table 10. First in sub-table (a), we gather negative sentence examples from other GPUs, significantly increasing the number of negatives, by a factor of number of GPUs, which in our case was 128. Such approach indeed helps reaching lower cross-lingual similarity search error rates. The increasing number of negative examples comes also at the price of higher probability of considering false negative sentences in the loss. We ablate the use of false negative removal heuristic presented in Section 4.3, and validate the usefulness of such approach.

Finally, in sub-table (b), we extend the in-batch negatives with the hard negatives presented in Section 4.4, either using a single contrastive learning task (one softmax) for both in-batch and hard negatives, or two contrastive learning tasks (split softmax). The first interesting finding is that training a model using hard negatives with a non-zero margin does not converge correctly. Therefore, we do not use any margin in the "one softmax" setup. This leads us to use $m = 0.3$ for in-batch negatives and $m = 0$ for hard negatives in the "split softmax" setup. We notice that hard negatives significantly lower xsim++ error rates. However,

not separating the hard negatives from in-batch negatives in two different contrastive loss terms affects xsim performance. This highlights the benefits of having two contrastive learning losses, one for in-batch negatives and another for hard negatives, to better balance the two in the final loss.

## C.3 Model Initialization

In order to understand the benefits of initializing from LLaMA, we ablate starting the seq2seq stage both from LLaMA, and from random initialization. This analysis is present in Table 11a. It is possible to see that initializing from LLaMA brings large improvements over random initialization in both spBLEU and chrF++, despite officially only supporting 8 languages, performing this extension to 200 languages is still easier than training from scratch.

To understand the advantage of doing a first seq2seq step to adapt LLaMA to many languages and give it the ability to encode and decode, we initialize our contrastive step from LLaMA, Seq2Seq and also random. Those results are available in Table 11b, where it is possible to see the very large improvements in xsim and xsim++ obtained from starting from the seq2seq model instead of from LLaMA.

## C.4 Data Mixes

Table 12 presents an ablation study evaluating the impact of various data processing steps on model performance, as measured by the xsim and xsim++ metrics. Starting with the baseline NLLB data, we incrementally apply different data modifications: filtering, addition of code/math content, and removal of false negatives. For each configuration, we report the resulting xsim and xsim++ scores. Both filtering and the addition of code and math seem to bring small beneficial changes, but a large improvement is seen in false negative removal, suggesting that even more aggressive filtering in the data could lead to further improvements.

## C.5 Pooling

It is a common debate whether to use mean pooling or CLS pooling, with SONAR (Duquenne et al., 2023) reporting better result with mean pooling, while MEXMA (Janeiro et al., 2025b) reported better results with CLS pooling. Intuitively, CLS pooling should work better, since it has the freedom to attend differently to each tokens. In Table 13 we experiment with both pooling methods and find that our model performs best with CLS pooling.

| Initialization | spBLEU | chrF++ |
|---|---|---|
| Random | 17.22 | 36.55 |
| LLaMA | 23.57 | 42.71 |

(a) Ablation on model initialization for the Seq2Seq stage.

| Initialization | xsim | xsim++ |
|---|---|---|
| Random init. | 13.35 | 71.30 |
| LLaMA init. | 1.02 | 11.98 |
| Seq2Seq init. | 0.65 | 8.95 |

(b) Ablation on model initialization for Contrastive Learning stage.

Table 11: **Model initialization ablations**: Effect of model weight initialization for sequence-to-sequence stage as well as for contrastive learning stage on respectively decoding performance (spBLEU and chrF++) and cross-lingual similariy search (xsim and xsim++) on FLORES200 dev set.

| Data | xsim | xsim++ |
|---|---|---|
| NLLB data | 0.74 | 10.03 |
| + filtering | 0.71 | 9.62 |
| + code/math | 0.70 | 9.50 |
| + false negatives removal | 0.65 | 8.95 |

Table 12: **Ablations on data** Ablation on the datamix used for contrastive finetuning on cross-lingual similariy search on FLORES200 dev set.

### C.6 SMALLER SCALE MODELS

To make our models more accessible to practitioners with varying computational constraints, we investigate the performance of smaller-scale variants of ECHO. A key design goal is ensuring these smaller models serve as drop-in replacements across all scales, enabling practitioners to seamlessly switch between model sizes while maintaining compatibility with downstream components.

**Model Pruning Strategy.** We create smaller models through structured pruning of the original 1.5B parameter model. Our pruning approach encompasses multiple architectural dimensions: (1) reducing inner model dimensionality (from 2048 to 1024-1792), (2) decreasing the number of encoder layers (from 16 to 8-14), (3) adjusting attention heads proportionally, and (4) scaling the feed-forward network dimensions accordingly. For layer selection, we employ a strategic sampling approach that preserves both the first and last layers while uniformly sampling intermediate layers, maintaining representational capacity across network depth.

**Knowledge Distillation.** Rather than training smaller models from scratch, we leverage knowledge distillation to ensure all model variants produce representations in the same aligned embedding space. We use the full 1.5B parameter ECHO model as the teacher and train smaller student models using Mean Squared Error (MSE) loss on the output embeddings. This approach offers a critical advantage: any task-specific decoder or classifier trained on representations from one model size can be directly applied to representations from any other size, as all models produce semantically aligned embeddings in the same 1024-dimensional space. This design enables practitioners to optimize their compute-performance trade-off dynamically. A user might develop and evaluate with the large model, then deploy a smaller variant for production inference, or vice versa, training efficiently on a smaller model and scaling up for final deployment, all while maintaining full compatibility with existing task-specific components.

In Table 14, we demonstrate that even our smallest model (Tiny, 385M parameters) retains approximately 76% of the full model's performance on cross-lingual similarity search, as measured by relative xsim++ scores. The Medium (1.1B) and Small (806M) models show even stronger performance, achieving over 85% of the full model's capability while offering substantial computational savings. Importantly, all models maintain strong cross-lingual alignment across all 200 supported languages, with minimal performance degradation on the 80 languages common across baseline models.

| Model | xsim | xsim++ |
|---|---|---|
| Mean | 0.68 | 9.25 |
| CLS | 0.64 | 8.77 |

Table 13: Ablation on different pooling strategies, evaluated on FLORES200 dev set.

| Model | Size | xsim (all) | xsim++ (all) | xsim (common) | xsim++ (common) |
|-------|------|-----------|--------------|---------------|-----------------|
| ECHO (Large) | 1.5B | 0.99 | 8.12 | 0.07 | 3.9 |
| Medium | 1.1B | 1.18 | 8.78 | 0.07 | 4.21 |
| Small | 806M | 1.23 | 9.01 | 0.08 | 4.23 |
| Tiny | 385M | 1.61 | 11.71 | 0.09 | 5.13 |

Table 14: Results of smaller models in xsim, and xsim++. "Common" refers to the 80 languages aforementioned, and "all" to all 200 languages covered by our model.

| model | std | mean |
|-------|-----|------|
| MEXMA | 0.0312 | -0.0011 |
| mE5$_{large}$ | 0.0312 | -0.0008 |
| LaBSE | 0.0358 | 0.0049 |
| SONAR | 0.0074 | 0.0000 |
| ECHO | 0.0356 | -0.0006 |

Table 15: Standard deviation (std) and mean of embedding features for different models when encoding FLORES200 dev set on all common languages.

## C.7 MODEL REPRESENTATION COLLAPSE

An often overlooked aspect of learned representations is how much of the embedding space they actually utilize, that is, whether their representations are collapsed within the space. Duquenne et al. (2023) have already highlighted this issue, which is especially pronounced when training with MSE regression signals, as models may exploit collapse to minimize the loss. This issue is crucial in the deployment of embeddings in current production systems that leverage mixed precision to reduce the memory footprint, as collapse can largely affect performance at lower precision. In Table 15 we see how ECHO successfully avoids collapse compared to other models like SONAR, with a healthy standard deviation on its features, similar to widely used models such as mE5$_{large}$.

## C.8 EMBEDDING DIMENSION INFORMATIVENESS

Singular Value Decomposition (SVD) provides a principled approach to analyze the intrinsic dimensionality and information distribution in embeddings. By examining the decay pattern of singular values, we can assess how different models utilize their feature space and identify potential dimensional collapse, where models concentrate information in fewer dimensions than their nominal embedding size.

Figure 6 plots the SVD of our baselines on the FLORES dev set. From it, it can be inferred that ECHO showcases a stable decay pattern reaching up to 800 dimensions, while other models decay earlier, with the sole exception of SONAR.

## C.9 ANALYZING EXAMPLES TO UNDERSTAND WHERE MODELS FAIL

In this section, we perform some qualitative analysis of the errors of ECHO, and other models. By inspection, ECHO's mistakes look to be related to unit conversion, matching with the sentence where the actual number matches, i.e. matching "15 cm" to "15 inches" instead of "6 inches". This is likely due to our hard negatives, which focused on matching the actual numbers, but lead to errors when the translation transforms the units.

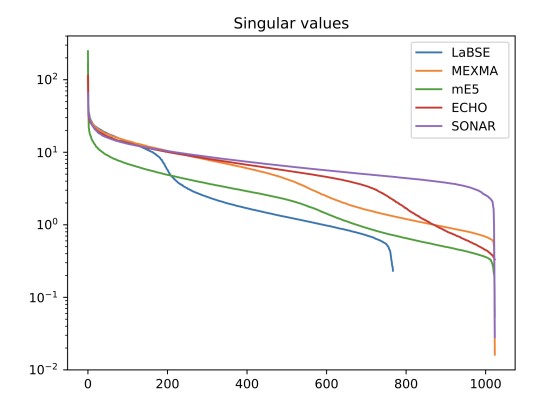

Figure 6: Singular values of embeddings from different models.

Meanwhile we see SONAR and MEXMA make mistakes related to both values and semantics (may/will, white/black), such as the examples provided below. Examples are provided in Table 16.

## D    TOKENIZER TRAINING

To extend the tokenizer vocabulary, we implemented a byte-pair encoding "continued training" algorithm by sequentially merging the most frequently occurring consecutive pairs of tokens within a word. The word frequencies were computed with a balanced sample from the parallel training data in all our languages and from the FineWeb2 dataset of web documents (in equal proportions). As weights for balancing, we used the total number of characters in the texts, and we applied unimax sampling over the languages, squashing the proportions of the first 126 languages to uniform and upsampling the rest at most x100 (on top of this, we manually increased the weights for some languages with underrepresented scripts, such as Greek or Korean, to adjust the resulting tokenizer fertilities). For some languages, the bottleneck of tokenization fertility has been not in the vocabulary itself but in the pre-tokenization word splitting regular expression, so we extended it with additional Unicode ranges and with a pattern for matching diacritic marks within a word. As a result of these operations, the extended tokenizer achieved the average fertility of 44 tokens per sentence over the 200 languages in the FLORES dataset, as opposed to 79 tokens in the original Llama3 tokenizer.[2]

## E    PROMPTS

Table Table 17 shows the prompts we used when tokenizing the input for both the Encoder and the Decoder, as explained in Section 4.2.

## F    FULL RESULTS

We present a breakdown of the cross-lingual similarity search results for our 200 focus languages in Table 18 and Table 19.

---

[2]With the most pronounced differences for the Asian languages with unique scripts, such as shn_Mymr, sat_Olck, and dzo_Tibt, where the fertility has decreased by more than 6 times.

| System | Source Sentence | Desired Retrieved | Actual Retrieved |
|--------|-----------------|-------------------|------------------|
| ECHO | O Corpo de Engenheiros dos EUA estimou que 15 cm de chuva podem romper os diques anteriormente danificados. | The U.S. Corps of Engineers estimated that 6 inches of rainfall could breach the previously damaged levees. | The U.S. Corps of Engineers estimated that 15 inches of rainfall could breach the previously damaged levees. |
| ECHO | Os limites de velocidade anunciados sao visivelmente mais baixos do que nas secoes anteriores e subsequentes - comumente 55-65 km/h - e a estrita obediencia a eles e ainda mais importante do que o contrario. | Posted speed limits are noticeably lower than in previous and subsequent sections commonly 35-40 mph (56-64 km/h) and strict obedience to them is even more important than otherwise. | Posted speed limits are noticeably lower than in previous and subsequent sections commonly 35-90 mph (56-64 km/h) and strict obedience to them is even more important than otherwise. |
| MEXMA | O Corpo de Engenheiros dos EUA estimou que 15 cm de chuva podem romper os diques anteriormente danificados. | The U.S. Corps of Engineers estimated that 6 inches of rainfall could breach the previously damaged levees. | The U.S. Corps of Engineers estimated that 15 inches of rainfall could breach the previously damaged levees. |
| MEXMA | Reportagens televisivas divulgam a fumaca esbranquicada saindo da planta. | Television reports show white smoke coming from the plant. | Television reports show black smoke coming from the plant. |
| SONAR | No periodo de um ano, uma pessoa infectada pode infectar entre 10 e 15 contatos proximos. | In one year's time, an infected person may infect 10 to 15 close contacts. | In one year's time, an infected person will infect 10 to 15 close contacts. |
| SONAR | Aconteceu novamente no mesmo mes em Mashhad, outro aviao comercial entrou em uma pista e atingiu uma parede, matando dezessete pessoas. | The same month saw another airliner overrun a runway at Mashhad and strike a wall, killing seventeen. | The same month did not saw another airliner overrun a runway at Mashhad and strike a wall, killing seventeen. |

Table 16: Comparison of three systems (ECHO, MEXMA, SONAR) on two examples each. For each example, the table shows the original source sentence (in Portuguese) and the desired retrieved English sentence, the actual retrieved English sentence.

| Source | Prompt Template |
|--------|-----------------|
| **Encoder** | |
| Source/Anchor | `"<CLS> [LANGUAGE]:<SEP> [INPUT SENTENCE] "` |
| **Decoder** | |
| NLLB Primary | `" This is a possible translation in [LANGUAGE]:<SEP> [INPUT SENTENCE] "` |
| NLLB Mined | `" This is a possible mined translation in [LANGUAGE]:<SEP> [INPUT SENTENCE] "` |
| NLLB *_bt | `" This is a possible back-translation in [LANGUAGE]:<SEP> [INPUT SENTENCE] "` |
| Eng → Code | `" This is a corresponding code snippet in [LANGUAGE]:<SEP> [INPUT SENTENCE] "` |
| Eng → Math | `" This is a corresponding math formula:<SEP> [INPUT SENTENCE] "` |
| Code/Math → Eng | `" This is a possible natural language explanation in [LANGUAGE]:<SEP> [INPUT SENTENCE] "` |

Table 17: Prompt Templates for Encoder and Decoder Components. The encoder uses classification prompts to identify language and content, while the decoder uses descriptive prompts tailored to different data sources and translation types. Placeholders `[LANGUAGE]` and `[INPUT SENTENCE]` are replaced with actual values during training.

| Lang | SONAR | LaBSE | MEXMA | ECHO | mE5 | Lang | SONAR | LaBSE | MEXMA | ECHO | mE5 | Lang | SONAR | LaBSE | MEXMA | ECHO | mE5 | Lang | SONAR | LaBSE | MEXMA | ECHO | mE5 |
|---|---|---|---|---|---|---|---|---|---|---|---|---|---|---|---|---|---|---|---|---|---|---|---|
| ace_Arab | 6.23 | 96.44 | 57.91 | 8.60 | 77.77 | ace_Latn | 0.30 | 32.61 | 8.60 | 0.10 | 8.70 | acm_Arab | 0.10 | 0.30 | 0.00 | 0.00 | 0.10 | acq_Arab | 0.00 | 0.10 | 0.00 | 0.00 | 0.10 |
| aeb_Arab | 0.40 | 4.94 | 0.20 | 0.20 | 0.99 | afr_Latn | 0.00 | 0.00 | 0.00 | 0.00 | 0.00 | ajp_Arab | 0.10 | 0.49 | 0.00 | 0.00 | 0.20 | aka_Latn | 0.30 | 53.46 | 44.86 | 0.10 | 7.21 |
| als_Latn | 0.00 | 0.00 | 0.00 | 0.00 | 0.00 | amh_Ethi | 0.00 | 0.00 | 0.00 | 0.00 | 0.40 | apc_Arab | 0.10 | 1.09 | 0.00 | 0.00 | 0.10 | arb_Arab | 0.00 | 0.00 | 0.00 | 0.00 | 0.00 |
| ars_Arab | 0.10 | 0.00 | 0.00 | 0.00 | 0.10 | ary_Arab | 0.99 | 13.14 | 0.99 | 0.89 | 2.77 | arz_Arab | 0.20 | 0.69 | 0.10 | 0.00 | 0.40 | asm_Beng | 0.10 | 1.78 | 0.00 | 0.00 | 0.69 |
| ast_Latn | 0.00 | 0.20 | 0.00 | 0.00 | 0.00 | awa_Deva | 0.99 | 1.09 | 0.89 | 0.89 | 0.99 | ayr_Latn | 3.85 | 72.13 | 54.25 | 1.68 | 43.97 | azb_Arab | 2.96 | 44.37 | 1.68 | 0.69 | 11.36 |
| azj_Latn | 0.30 | 0.30 | 0.20 | 0.20 | 0.20 | bak_Cyrl | 0.00 | 41.90 | 11.36 | 0.00 | 1.68 | bam_Latn | 4.05 | 65.61 | 52.37 | 2.17 | 14.62 | ban_Latn | 0.40 | 8.40 | 1.09 | 0.30 | 2.57 |
| bel_Cyrl | 0.49 | 0.00 | 0.00 | 0.00 | 0.20 | bem_Latn | 0.00 | 44.07 | 36.66 | 0.10 | 14.23 | ben_Beng | 0.00 | 0.00 | 0.00 | 0.00 | 0.10 | bho_Deva | 0.20 | 2.57 | 0.30 | 0.00 | 0.79 |
| bjn_Arab | 4.84 | 95.36 | 69.96 | 6.23 | 82.41 | bjn_Latn | 0.10 | 8.60 | 0.30 | 0.10 | 1.68 | bod_Tibt | 1.28 | 14.13 | 88.93 | 0.49 | 92.19 | bos_Latn | 0.00 | 0.00 | 0.00 | 0.00 | 0.00 |
| bug_Latn | 0.79 | 40.61 | 13.14 | 0.49 | 12.65 | bul_Cyrl | 0.10 | 0.00 | 0.00 | 0.00 | 0.10 | cat_Latn | 0.00 | 0.00 | 0.00 | 0.00 | 0.00 | ceb_Latn | 0.00 | 0.00 | 6.82 | 0.00 | 0.10 |
| ces_Latn | 0.00 | 0.00 | 0.00 | 0.00 | 0.00 | cjk_Latn | 12.55 | 62.45 | 42.69 | 7.02 | 43.97 | ckb_Arab | 0.10 | 88.83 | 0.10 | 0.00 | 3.26 | crh_Latn | 0.10 | 2.67 | 0.10 | 0.00 | 0.30 |
| cym_Latn | 0.00 | 0.00 | 0.00 | 0.00 | 0.00 | dan_Latn | 0.00 | 0.00 | 0.00 | 0.00 | 0.49 | deu_Latn | 0.00 | 0.00 | 0.00 | 0.00 | 0.00 | dik_Latn | 11.26 | 62.25 | 46.15 | 8.89 | 46.34 |
| dyu_Latn | 21.34 | 74.51 | 53.36 | 13.54 | 50.30 | dzo_Tibt | 1.19 | 67.19 | 99.41 | 0.49 | 99.51 | ell_Grek | 0.00 | 0.00 | 0.00 | 0.00 | 0.00 | epo_Latn | 0.00 | 0.00 | 0.00 | 0.00 | 0.10 |
| est_Latn | 0.00 | 0.00 | 0.00 | 0.00 | 0.10 | eus_Latn | 0.00 | 0.10 | 0.00 | 0.00 | 0.00 | ewe_Latn | 1.19 | 64.53 | 53.16 | 0.89 | 16.21 | fao_Latn | 0.10 | 0.49 | 0.00 | 0.00 | 2.47 |
| fij_Latn | 0.49 | 60.77 | 52.27 | 0.30 | 13.24 | fin_Latn | 0.10 | 0.10 | 0.10 | 0.10 | 0.10 | fon_Latn | 5.83 | 70.16 | 57.41 | 4.64 | 19.66 | fra_Latn | 0.30 | 0.00 | 0.00 | 0.00 | 0.00 |
| fur_Latn | 0.00 | 12.06 | 0.20 | 0.00 | 0.89 | fuv_Latn | 10.97 | 65.02 | 43.38 | 4.55 | 36.86 | gaz_Latn | 0.20 | 81.72 | 47.92 | 0.10 | 12.15 | gla_Latn | 0.10 | 0.20 | 0.10 | 0.10 | 4.25 |
| gle_Latn | 0.00 | 0.00 | 0.00 | 0.00 | 1.19 | glg_Latn | 0.00 | 0.00 | 0.00 | 0.00 | 0.00 | grn_Latn | 0.30 | 47.92 | 27.37 | 0.40 | 10.87 | guj_Gujr | 0.00 | 0.00 | 0.00 | 0.00 | 0.00 |
| hat_Latn | 0.59 | 0.59 | 13.83 | 0.59 | 1.28 | hau_Latn | 0.40 | 0.30 | 0.30 | 0.30 | 2.67 | heb_Hebr | 0.00 | 0.00 | 0.00 | 0.00 | 0.00 | hin_Deva | 0.00 | 0.00 | 0.00 | 0.00 | 0.00 |
| hne_Deva | 0.40 | 1.78 | 0.40 | 0.00 | 0.69 | hrv_Latn | 0.00 | 0.00 | 0.00 | 0.00 | 0.00 | hun_Latn | 0.10 | 0.00 | 0.00 | 0.00 | 0.00 | hye_Armn | 0.00 | 0.00 | 0.00 | 0.00 | 0.00 |
| ibo_Latn | 0.10 | 1.09 | 48.81 | 0.00 | 4.45 | ilo_Latn | 0.00 | 30.24 | 16.30 | 0.00 | 1.68 | ind_Latn | 0.00 | 0.00 | 0.00 | 0.00 | 0.30 | isl_Latn | 0.20 | 0.10 | 0.10 | 0.10 | 0.10 |
| ita_Latn | 0.20 | 0.00 | 0.00 | 0.00 | 0.20 | jav_Latn | 0.10 | 0.00 | 0.00 | 0.00 | 0.00 | jpn_Jpan | 0.20 | 0.00 | 0.10 | 0.00 | 0.00 | kab_Latn | 0.10 | 82.41 | 67.19 | 0.00 | 37.35 |
| kac_Latn | 1.78 | 67.98 | 51.09 | 0.10 | 41.40 | kam_Latn | 3.36 | 54.45 | 38.74 | 2.17 | 29.25 | kan_Knda | 0.00 | 0.00 | 0.00 | 0.00 | 0.30 | kas_Arab | 0.20 | 34.88 | 3.06 | 0.20 | 4.84 |
| kas_Deva | 1.88 | 56.72 | 15.91 | 0.59 | 16.60 | kat_Geor | 0.40 | 0.00 | 0.00 | 0.00 | 0.10 | kaz_Cyrl | 0.30 | 0.20 | 0.20 | 0.20 | 0.30 | kbp_Latn | 4.94 | 67.79 | 55.34 | 4.35 | 39.33 |
| kea_Latn | 0.00 | 14.82 | 1.19 | 0.00 | 0.79 | khk_Cyrl | 0.30 | 0.00 | 0.10 | 0.00 | 0.59 | khm_Khmr | 0.00 | 2.37 | 0.00 | 0.69 | 0.79 | kik_Latn | 0.89 | 52.37 | 43.18 | 0.59 | 6.72 |
| kin_Latn | 0.30 | 0.30 | 49.51 | 0.20 | 2.87 | kir_Cyrl | 0.30 | 0.10 | 0.00 | 0.00 | 0.59 | kmb_Latn | 0.89 | 61.66 | 48.02 | 1.28 | 36.76 | kmr_Latn | 0.20 | 0.30 | 3.66 | 0.00 | 2.17 |
| knc_Arab | 63.74 | 96.74 | 80.14 | 50.89 | 79.55 | knc_Latn | 7.81 | 65.22 | 42.39 | 0.99 | 45.45 | kon_Latn | 0.40 | 52.47 | 40.42 | 0.20 | 9.29 | kor_Hang | 0.10 | 0.00 | 0.00 | 0.00 | 0.20 |
| lao_Laoo | 0.00 | 3.46 | 0.00 | 0.00 | 0.79 | lij_Latn | 0.10 | 10.57 | 0.59 | 0.10 | 1.38 | lim_Latn | 0.20 | 9.09 | 0.30 | 0.00 | 3.56 | lin_Latn | 0.20 | 50.69 | 40.71 | 0.20 | 3.85 |
| lit_Latn | 0.49 | 0.40 | 0.49 | 0.40 | 0.40 | lmo_Latn | 0.30 | 16.40 | 0.69 | 0.00 | 2.77 | ltg_Latn | 0.10 | 25.20 | 12.65 | 0.10 | 5.34 | ltz_Latn | 0.00 | 0.00 | 4.55 | 0.00 | 0.89 |
| lua_Latn | 1.28 | 50.49 | 38.04 | 0.49 | 16.80 | lug_Latn | 0.20 | 45.65 | 41.90 | 0.30 | 9.78 | luo_Latn | 0.00 | 64.43 | 49.70 | 0.10 | 23.91 | lus_Latn | 1.48 | 52.47 | 36.36 | 0.49 | 15.81 |
| lvs_Latn | 0.20 | 0.00 | 0.00 | 0.00 | 0.00 | mag_Deva | 0.10 | 0.30 | 0.00 | 0.10 | 0.00 | mai_Deva | 0.00 | 0.20 | 0.10 | 0.00 | 0.10 | mal_Mlym | 0.10 | 0.10 | 0.10 | 0.10 | 0.10 |
| mar_Deva | 0.00 | 0.00 | 0.00 | 0.00 | 0.10 | min_Latn | 0.10 | 12.85 | 0.89 | 0.10 | 1.98 | mkd_Cyrl | 0.00 | 0.00 | 0.00 | 0.00 | 0.00 | mlt_Latn | 0.00 | 0.00 | 15.71 | 0.00 | 0.79 |
| mni_Beng | 0.00 | 90.02 | 72.13 | 0.30 | 46.84 | mos_Latn | 10.67 | 70.36 | 51.19 | 5.73 | 45.16 | mri_Latn | 0.10 | 2.47 | 57.91 | 0.00 | 11.56 | mya_Mymr | 0.69 | 0.30 | 0.20 | 0.20 | 0.69 |
| nld_Latn | 0.40 | 0.00 | 0.00 | 0.00 | 0.00 | nob_Latn | 0.10 | 0.10 | 0.10 | 0.10 | 0.10 | nno_Latn | 0.20 | 0.10 | 0.10 | 0.10 | 0.10 | npi_Deva | 0.59 | 0.30 | 0.30 | 0.30 | 0.40 |
| nso_Latn | 0.10 | 7.02 | 44.66 | 0.10 | 2.96 | nus_Latn | 2.27 | 79.45 | 64.92 | 1.98 | 49.41 | nya_Latn | 0.10 | 0.79 | 37.55 | 0.20 | 3.85 | oci_Latn | 0.00 | 0.00 | 0.49 | 0.10 | 0.10 |
| ory_Orya | 0.20 | 0.00 | 0.00 | 0.00 | 0.10 | pag_Latn | 0.89 | 30.43 | 17.39 | 0.30 | 4.35 | pan_Guru | 0.00 | 0.00 | 0.00 | 0.00 | 0.10 | pap_Latn | 0.00 | 11.26 | 1.09 | 0.00 | 0.30 |
| pbt_Arab | 0.10 | 1.09 | 0.10 | 0.00 | 0.49 | pes_Arab | 0.20 | 0.00 | 0.00 | 0.00 | 0.00 | plt_Latn | 0.00 | 0.49 | 15.32 | 0.00 | 1.19 | pol_Latn | 0.00 | 0.00 | 0.00 | 0.00 | 0.20 |
| por_Latn | 0.00 | 0.00 | 0.00 | 0.00 | 0.59 | prs_Arab | 0.10 | 0.00 | 0.00 | 0.00 | 0.10 | quy_Latn | 3.95 | 67.39 | 42.49 | 3.16 | 30.83 | ron_Latn | 0.00 | 0.00 | 0.00 | 0.00 | 0.00 |
| run_Latn | 0.20 | 2.87 | 48.32 | 0.10 | 3.85 | rus_Cyrl | 0.20 | 0.00 | 0.00 | 0.00 | 0.00 | sag_Latn | 3.16 | 60.97 | 43.28 | 1.78 | 33.89 | san_Deva | 0.69 | 19.17 | 0.40 | 0.40 | 2.08 |
| scn_Latn | 0.30 | 8.30 | 1.09 | 0.00 | 1.68 | shn_Mymr | 0.49 | 71.54 | 53.06 | 0.00 | 42.19 | sin_Sinh | 0.30 | 0.00 | 0.00 | 0.10 | 0.00 | slk_Latn | 0.10 | 0.00 | 0.00 | 0.00 | 0.00 |
| slv_Latn | 0.10 | 0.00 | 0.00 | 0.00 | 0.10 | smo_Latn | 0.10 | 1.38 | 49.41 | 0.10 | 4.55 | sna_Latn | 0.20 | 2.37 | 43.18 | 0.20 | 3.16 | snd_Arab | 0.00 | 0.00 | 0.00 | 0.00 | 0.49 |
| som_Latn | 0.10 | 1.09 | 0.10 | 0.10 | 4.64 | sot_Latn | 0.00 | 0.59 | 46.94 | 0.00 | 1.78 | spa_Latn | 0.10 | 0.10 | 0.10 | 0.10 | 0.10 | srd_Latn | 0.00 | 9.09 | 0.49 | 0.00 | 0.79 |
| srp_Cyrl | 0.00 | 0.00 | 0.00 | 0.00 | 0.00 | ssw_Latn | 0.49 | 16.70 | 6.32 | 0.30 | 6.52 | tam_Taml | 0.10 | 0.00 | 0.10 | 0.00 | 0.10 | taq_Latn | 22.33 | 66.80 | 48.81 | 16.90 | 48.42 |
| swh_Latn | 0.00 | 0.00 | 0.00 | 0.00 | 0.69 | szl_Latn | 0.69 | 4.94 | 0.79 | 0.69 | 0.79 | tel_Telu | 0.20 | 0.00 | 0.00 | 0.00 | 0.00 | tgk_Cyrl | 0.20 | 0.30 | 49.01 | 0.20 | 1.48 |
| taq_Tfng | 21.34 | 95.55 | 86.07 | 25.79 | 87.55 | tat_Cyrl | 0.00 | 0.00 | 5.83 | 0.00 | 0.59 | tir_Ethi | 0.40 | 77.27 | 16.70 | 0.00 | 6.32 | tpi_Latn | 0.00 | 46.84 | 17.39 | 0.00 | 3.36 |
| tgl_Latn | 0.00 | 0.00 | 0.30 | 0.00 | 0.00 | tha_Thai | 0.10 | 6.62 | 0.10 | 0.10 | 0.40 | tuk_Latn | 0.10 | 0.69 | 5.04 | 0.00 | 19.47 | tum_Latn | 0.79 | 24.21 | 41.40 | 0.20 | 6.13 |
| tsn_Latn | 1.09 | 8.50 | 51.58 | 1.09 | 4.25 | tso_Latn | 0.49 | 55.34 | 43.08 | 0.40 | 5.14 | tzm_Tfng | 0.79 | 95.55 | 89.43 | 0.99 | 90.42 | uig_Arab | 0.40 | 0.20 | 0.10 | 0.10 | 2.47 |
| tur_Latn | 0.00 | 0.00 | 0.00 | 0.00 | 0.00 | twi_Latn | 0.40 | 49.60 | 42.49 | 0.10 | 8.00 | urd_Arab | 0.20 | 0.10 | 0.10 | 0.10 | 0.40 | uzn_Latn | 0.10 | 0.10 | 0.10 | 0.00 | 0.00 |
| ukr_Cyrl | 0.00 | 0.00 | 0.00 | 0.00 | 0.00 | umb_Latn | 5.43 | 64.82 | 46.34 | 4.64 | 38.04 | war_Latn | 0.10 | 0.49 | 6.52 | 0.00 | 0.20 | wol_Latn | 0.99 | 54.84 | 42.09 | 1.09 | 17.19 |
| vec_Latn | 0.00 | 4.15 | 0.10 | 0.00 | 0.59 | vie_Latn | 0.10 | 0.00 | 0.00 | 0.00 | 0.10 | yor_Latn | 0.20 | 13.14 | 51.88 | 0.00 | 11.76 | yue_Hant | 0.20 | 0.10 | 0.00 | 0.00 | 0.00 |
| xho_Latn | 0.10 | 0.99 | 0.10 | 0.10 | 1.98 | ydd_Hebr | 0.00 | 0.89 | 0.20 | 0.00 | 2.87 | zsm_Latn | 0.00 | 0.00 | 0.00 | 0.00 | 0.10 | zul_Latn | 0.10 | 0.20 | 0.30 | 0.10 | 1.38 |
| zho_Hans | 0.00 | 0.00 | 0.10 | 0.00 | 0.00 | zho_Hant | 0.30 | 0.40 | 0.10 | 0.00 | 0.20 | | | | | | | | | | | | |

Table 18: xsim results for all models in all languages, x-eng in FLORES200 devtest set.

| Lang | SONAR | LaBSE | MEXMA | ECHO | mE5 | Lang | SONAR | LaBSE | MEXMA | ECHO | mE5 | Lang | SONAR | LaBSE | MEXMA | ECHO | mE5 | Lang | SONAR | LaBSE | MEXMA | ECHO | mE5 |
|---|---|---|---|---|---|---|---|---|---|---|---|---|---|---|---|---|---|---|---|---|---|---|---|
| ace_Arab | 55.34 | 100.00 | 92.09 | 36.76 | 98.91 | ace_Latn | 16.30 | 82.21 | 49.41 | 8.10 | 48.22 | acm_Arab | 11.17 | 52.67 | 9.39 | 5.83 | 27.37 | acq_Arab | 8.30 | 46.54 | 8.20 | 10.87 | 23.02 |
| aeb_Arab | 11.76 | 66.60 | 13.64 | 6.72 | 31.72 | afr_Latn | 5.14 | 9.49 | 4.64 | 1.19 | 13.34 | ajp_Arab | 7.61 | 54.15 | 9.19 | 5.24 | 23.81 | aka_Latn | 19.47 | 92.39 | 85.18 | 10.77 | 42.79 |
| als_Latn | 5.14 | 10.97 | 9.39 | 1.78 | 18.48 | amh_Ethi | 8.60 | 28.85 | 6.23 | 3.36 | 43.48 | apc_Arab | 9.58 | 58.00 | 12.65 | 5.14 | 28.46 | arb_Arab | 6.82 | 35.87 | 6.72 | 2.37 | 19.27 |
| ars_Arab | 13.93 | 39.72 | 12.65 | 8.50 | 23.52 | ary_Arab | 14.03 | 77.08 | 22.23 | 12.45 | 39.03 | arz_Arab | 10.87 | 58.99 | 11.46 | 4.84 | 24.51 | asm_Beng | 14.92 | 62.15 | 11.17 | 6.42 | 41.50 |
| ast_Latn | 9.68 | 38.04 | 14.13 | 6.82 | 19.86 | awa_Deva | 11.07 | 26.98 | 12.06 | 3.75 | 27.67 | ayr_Latn | 34.78 | 97.43 | 83.10 | 19.27 | 85.77 | azb_Arab | 42.59 | 94.76 | 32.61 | 21.25 | 65.32 |
| azj_Latn | 14.03 | 17.98 | 10.77 | 6.13 | 29.45 | bak_Cyrl | 11.46 | 91.90 | 51.09 | 3.56 | 40.71 | bam_Latn | 30.14 | 95.26 | 85.67 | 17.09 | 61.56 | ban_Latn | 13.04 | 60.38 | 29.15 | 6.13 | 36.26 |
| bel_Cyrl | 17.19 | 26.88 | 11.36 | 5.53 | 24.90 | bem_Latn | 14.82 | 87.35 | 73.02 | 6.92 | 54.05 | ben_Beng | 10.77 | 19.76 | 6.92 | 5.34 | 26.09 | bho_Deva | 12.45 | 45.95 | 20.45 | 4.94 | 36.86 |
| bjn_Arab | 42.69 | 100.00 | 95.06 | 31.13 | 98.62 | bjn_Latn | 11.76 | 58.30 | 20.06 | 4.94 | 33.20 | bod_Tibt | 25.99 | 82.51 | 97.23 | 17.89 | 99.21 | bos_Latn | 5.93 | 9.29 | 3.56 | 1.19 | 11.66 |
| bug_Latn | 24.11 | 85.57 | 52.27 | 12.25 | 56.62 | bul_Cyrl | 7.81 | 9.88 | 4.74 | 2.08 | 9.68 | cat_Latn | 4.84 | 12.94 | 3.95 | 1.88 | 8.10 | ceb_Latn | 9.29 | 16.60 | 48.42 | 3.16 | 26.19 |
| ces_Latn | 7.02 | 15.32 | 5.04 | 1.98 | 9.88 | cjk_Latn | 63.04 | 95.65 | 83.50 | 31.72 | 86.17 | ckb_Arab | 10.97 | 99.31 | 11.36 | 4.64 | 53.75 | crh_Latn | 9.19 | 50.59 | 21.64 | 3.95 | 37.94 |
| cym_Latn | 5.34 | 12.94 | 3.85 | 0.99 | 31.03 | dan_Latn | 4.84 | 7.21 | 3.75 | 1.09 | 9.78 | deu_Latn | 4.84 | 7.61 | 4.64 | 1.58 | 7.41 | dik_Latn | 46.94 | 94.66 | 79.05 | 34.19 | 79.74 |
| dyu_Latn | 65.51 | 97.83 | 83.79 | 41.21 | 86.66 | dzo_Tibt | 24.31 | 98.42 | 99.80 | 15.71 | 99.90 | ell_Grek | 9.19 | 18.77 | 7.41 | 2.87 | 13.14 | epo_Latn | 4.55 | 8.79 | 4.15 | 1.38 | 18.77 |
| est_Latn | 6.82 | 11.56 | 4.05 | 2.08 | 12.45 | eus_Latn | 9.88 | 14.13 | 7.11 | 2.96 | 20.45 | ewe_Latn | 22.63 | 96.34 | 83.40 | 13.83 | 59.98 | fao_Latn | 11.36 | 38.14 | 22.33 | 4.05 | 39.82 |
| fij_Latn | 16.01 | 94.66 | 84.39 | 8.30 | 53.95 | fin_Latn | 7.51 | 15.32 | 7.02 | 3.36 | 11.96 | fon_Latn | 35.08 | 96.05 | 87.15 | 26.38 | 62.85 | fra_Latn | 4.84 | 9.19 | 4.64 | 1.78 | 7.81 |
| fur_Latn | 5.83 | 71.05 | 25.59 | 4.45 | 34.09 | fuv_Latn | 49.51 | 96.15 | 81.62 | 27.57 | 76.38 | gaz_Latn | 16.30 | 98.72 | 83.10 | 8.70 | 60.47 | gla_Latn | 13.74 | 27.67 | 10.57 | 3.66 | 48.22 |
| gle_Latn | 8.70 | 17.49 | 8.70 | 3.26 | 39.03 | glg_Latn | 6.13 | 7.71 | 4.55 | 2.37 | 11.96 | grn_Latn | 18.87 | 91.50 | 68.08 | 9.58 | 55.83 | guj_Gujr | 8.50 | 15.02 | 6.62 | 3.06 | 31.72 |
| hat_Latn | 8.79 | 26.28 | 62.85 | 4.55 | 39.92 | hau_Latn | 11.26 | 28.16 | 11.36 | 5.14 | 37.94 | heb_Hebr | 5.43 | 17.00 | 6.52 | 2.77 | 18.68 | hin_Deva | 7.51 | 10.87 | 5.24 | 2.57 | 17.00 |
| hne_Deva | 9.58 | 39.92 | 16.21 | 4.15 | 31.52 | hrv_Latn | 7.02 | 9.88 | 4.64 | 2.96 | 13.24 | hun_Latn | 7.02 | 13.34 | 6.32 | 2.77 | 11.07 | hye_Armn | 6.32 | 11.86 | 6.92 | 2.67 | 32.51 |
| ibo_Latn | 12.06 | 45.95 | 79.84 | 6.52 | 43.28 | ilo_Latn | 10.18 | 82.81 | 55.93 | 4.64 | 35.28 | ind_Latn | 6.23 | 8.00 | 4.74 | 2.77 | 14.92 | isl_Latn | 8.50 | 14.43 | 6.72 | 3.46 | 19.86 |
| ita_Latn | 6.72 | 12.15 | 4.64 | 2.27 | 8.30 | jav_Latn | 8.79 | 19.07 | 7.81 | 3.85 | 23.91 | jpn_Jpan | 13.44 | 20.85 | 8.30 | 3.46 | 14.53 | kab_Latn | 22.23 | 98.81 | 93.08 | 17.00 | 86.46 |
| kac_Latn | 27.27 | 97.33 | 85.38 | 17.59 | 82.71 | kam_Latn | 34.98 | 92.98 | 74.90 | 22.92 | 72.73 | kan_Knda | 11.17 | 20.16 | 8.60 | 4.45 | 29.55 | kas_Arab | 16.90 | 90.61 | 45.45 | 9.88 | 52.17 |
| kas_Deva | 34.98 | 94.37 | 62.65 | 22.92 | 71.44 | kat_Geor | 12.94 | 24.11 | 8.89 | 4.74 | 32.51 | kaz_Cyrl | 10.77 | 13.83 | 7.41 | 3.95 | 29.55 | kbp_Latn | 29.15 | 95.65 | 89.53 | 18.18 | 83.50 |
| kea_Latn | 20.65 | 75.89 | 32.61 | 4.35 | 33.50 | khk_Cyrl | 13.34 | 24.01 | 17.29 | 5.83 | 38.54 | khm_Khmr | 11.86 | 21.11 | 8.00 | 8.20 | 45.45 | kik_Latn | 17.00 | 96.15 | 80.53 | 20.65 | 78.36 |
| kin_Latn | 9.49 | 29.64 | 82.41 | 4.35 | 36.36 | kir_Cyrl | 13.83 | 27.96 | 11.46 | 6.42 | 33.10 | kmb_Latn | 27.77 | 95.16 | 80.53 | 20.65 | 78.36 | kmr_Latn | 15.32 | 36.46 | 35.57 | 7.61 | 45.65 |
| knc_Arab | 89.82 | 100.00 | 95.55 | 77.57 | 97.33 | knc_Latn | 47.04 | 96.25 | 76.38 | 19.96 | 80.14 | kon_Latn | 17.69 | 93.58 | 76.78 | 10.57 | 50.20 | kor_Hang | 10.57 | 21.64 | 7.41 | 3.95 | 17.98 |
| lao_Laoo | 9.39 | 23.42 | 6.03 | 3.16 | 40.32 | lij_Latn | 8.79 | 68.38 | 24.90 | 3.66 | 30.73 | lim_Latn |  |  |  |  |  | lin_Latn |  |  |  |  |  |
| lit_Latn | 10.18 | 14.43 | 17.09 | 4.25 | 15.12 | lmo_Latn | 17.59 | 75.40 | 30.34 | 8.79 | 37.75 | ltg_Latn | 9.09 | 83.10 | 55.14 | 5.63 | 52.47 | ltz_Latn | 8.40 | 20.95 | 39.72 | 3.06 | 35.18 |
| lua_Latn | 32.51 | 91.60 | 75.40 | 16.50 | 59.78 | lug_Latn | 19.86 | 89.62 | 78.46 | 13.04 | 56.32 | luo_Latn | 12.65 | 95.75 | 83.30 | 7.31 | 65.81 | lus_Latn | 24.41 | 91.60 | 70.95 | 12.25 | 56.82 |
| lvs_Latn | 7.71 | 9.98 | 11.26 | 2.37 | 14.13 | mag_Deva | 8.70 | 32.51 | 16.60 | 4.25 | 28.66 | mai_Deva | 10.38 | 38.83 | 17.79 | 2.37 | 11.86 | mal_Mlym | 10.57 | 25.69 | 8.50 | 4.74 | 26.38 |
| mar_Deva | 9.29 | 18.87 | 6.42 | 3.66 | 26.19 | min_Latn | 9.49 | 64.53 | 23.52 | 3.85 | 36.46 | mkd_Cyrl | 6.62 | 9.98 | 5.14 | 2.27 | 11.86 | mlt_Latn | 5.04 | 8.89 | 60.08 | 1.68 | 27.77 |
| mni_Beng | 19.07 | 99.80 | 95.36 | 12.65 | 91.50 | mos_Latn | 41.60 | 96.64 | 83.30 | 26.38 | 86.07 | mri_Latn | 13.54 | 46.94 | 84.29 | 8.79 | 57.11 | mya_Mymr | 17.69 | 41.70 | 12.06 | 6.42 | 47.43 |
| nld_Latn | 10.87 | 12.55 | 7.11 | 3.56 | 9.78 | nno_Latn | 14.03 | 11.56 | 6.72 | 3.06 | 12.85 | nob_Latn | 11.76 | 10.18 | 6.72 | 3.06 | 12.85 | npi_Deva | 11.36 | 10.18 | 3.16 | 3.16 | 29.74 |
| nso_Latn | 9.88 | 59.98 | 75.99 | 4.64 | 36.26 | nus_Latn | 29.15 | 98.62 | 92.79 | 20.06 | 87.85 | nya_Latn | 13.34 | 43.58 | 71.64 | 7.91 | 36.17 | oci_Latn | 5.53 | 36.46 | 15.51 | 2.96 | 24.41 |
| ory_Orya | 9.78 | 18.18 | 10.77 | 2.67 | 28.75 | pag_Latn | 16.01 | 86.36 | 60.77 | 9.19 | 45.06 | pan_Guru | 9.58 | 21.54 | 11.46 | 3.06 | 31.03 | pap_Latn | 7.11 | 67.49 | 29.84 | 1.28 | 23.81 |
| pbt_Arab | 13.04 | 52.08 | 19.57 | 5.43 | 38.34 | pes_Arab | 8.70 | 11.86 | 6.42 | 2.77 | 16.60 | plt_Latn | 7.21 | 31.13 | 62.25 | 3.06 | 32.11 | pol_Latn | 8.70 | 12.35 | 6.03 | 3.16 | 9.19 |
| por_Latn | 5.43 | 8.20 | 5.04 | 1.68 | 17.59 | prs_Arab | 7.71 | 14.92 | 7.21 | 2.96 | 21.94 | quy_Latn | 28.85 | 96.15 | 80.43 | 17.29 | 77.08 | ron_Latn | 5.83 | 7.02 | 3.46 | 1.68 | 7.41 |
| run_Latn | 11.07 | 51.38 | 80.43 | 4.84 | 42.98 | rus_Cyrl | 6.52 | 10.28 | 6.13 | 2.57 | 9.88 | sag_Latn | 39.23 | 95.75 | 80.04 | 26.09 | 76.98 | san_Deva | 19.96 | 80.14 | 19.86 | 8.40 | 45.55 |
| scn_Latn | 12.25 | 62.85 | 33.30 | 6.42 | 41.11 | shn_Mymr | 18.97 | 96.44 | 76.58 | 10.18 | 83.60 | sin_Sinh | 9.09 | 18.18 | 6.13 | 4.25 | 34.78 | slk_Latn | 8.10 | 9.88 | 5.53 | 2.67 | 12.55 |
| slv_Latn | 7.91 | 12.75 | 5.34 | 2.27 | 13.14 | smo_Latn | 11.96 | 41.50 | 83.40 | 5.53 | 44.57 | sna_Latn | 11.76 | 49.11 | 77.27 | 4.05 | 40.91 | snd_Arab | 11.17 | 43.87 | 8.10 | 4.74 | 45.85 |
| som_Latn | 12.15 | 41.70 | 13.04 | 8.70 | 45.06 | sot_Latn | 7.91 | 43.18 | 79.64 | 4.35 | 34.98 | spa_Latn | 8.00 | 14.92 | 5.43 | 2.67 | 9.98 | srd_Latn | 10.47 | 66.70 | 26.78 | 6.13 | 33.79 |
| srp_Cyrl | 5.43 | 10.38 | 3.66 | 1.38 | 10.08 | ssw_Latn | 12.06 | 74.21 | 47.92 | 6.62 | 45.55 | sun_Latn |  |  |  |  |  | swe_Latn | 5.83 | 18.18 | 8.10 | 3.85 | 29.64 |
| swh_Latn | 7.11 | 16.80 | 7.71 | 2.77 | 28.75 | szl_Latn | 6.72 | 57.61 | 18.68 | 3.56 | 32.51 | tam_Taml | 14.23 | 18.68 | 9.29 | 4.05 | 31.92 | taq_Latn | 57.61 | 96.05 | 79.84 | 39.43 | 86.17 |
| taq_Tfng | 62.35 | 100.00 | 96.64 | 53.46 | 98.02 | tat_Cyrl | 7.91 | 23.62 | 43.18 | 3.46 | 38.34 | tel_Telu | 12.06 | 16.01 | 8.40 | 3.85 | 26.38 | tgk_Cyrl | 8.40 | 23.81 | 82.91 | 3.66 | 45.26 |
| tgl_Latn | 6.62 | 12.75 | 25.49 | 2.67 | 22.43 | tha_Thai | 8.30 | 39.43 | 6.23 | 3.06 | 14.33 | tir_Ethi | 14.82 | 98.52 | 64.62 | 7.11 | 58.00 | tpi_Latn | 13.64 | 98.47 | 91.56 | 7.91 | 42.98 |
| tsn_Latn | 13.54 | 61.07 | 82.71 | 6.03 | 40.91 | tso_Latn | 13.14 | 91.80 | 74.80 | 5.34 | 40.91 | tuk_Latn | 9.49 | 40.51 | 42.59 | 3.85 | 76.98 | tum_Latn | 18.28 | 78.06 | 73.12 | 9.68 | 44.07 |
| tur_Latn | 6.23 | 10.67 | 5.04 | 2.37 | 12.55 | twi_Latn | 18.28 | 91.60 | 85.47 | 9.68 | 45.75 | tzm_Tfng | 26.88 | 100.00 | 97.33 | 18.08 | 98.72 | uig_Arab | 13.83 | 28.56 | 11.07 | 6.82 | 54.25 |
| ukr_Cyrl | 7.91 | 11.96 | 6.42 | 3.16 | 10.08 | umb_Latn | 36.56 | 95.06 | 77.57 | 26.98 | 77.47 | urd_Arab | 9.88 | 17.79 | 6.42 | 4.15 | 30.43 | uzn_Latn | 10.87 | 16.60 | 6.03 | 3.16 | 9.19 |
| vec_Latn | 7.81 | 53.46 | 14.03 | 2.67 | 28.66 | vie_Latn | 5.63 | 11.56 | 5.53 | 2.27 | 12.06 | war_Latn | 7.11 | 32.51 | 47.04 | 2.87 | 24.60 | wol_Latn | 28.56 | 93.87 | 77.77 | 16.90 | 66.60 |
| xho_Latn | 10.18 | 41.01 | 12.94 | 4.74 | 34.68 | ydd_Hebr | 8.60 | 52.67 | 31.62 | 3.36 | 57.61 | yor_Latn | 22.73 | 71.25 | 84.49 | 16.80 | 58.79 | yue_Hant | 10.67 | 58.70 | 8.30 | 3.95 | 17.59 |
| zho_Hans | 9.98 | 50.69 | 7.41 | 3.06 | 14.43 | zho_Hant | 14.23 | 58.30 | 9.78 | 4.15 | 17.89 | zsm_Latn | 5.93 | 7.11 | 4.55 | 2.08 | 12.25 | zul_Latn | 8.70 | 33.10 | 21.64 | 4.05 | 34.49 |

Table 19: xsim++ results for all models in all languages, x-eng in FLORES200 devtest set.

## G EMBEDDING VISUALIZATION

So far, quantitative results have showcased the efficacy of ECHO as an embedding space. Although visualization approaches such as UMAP McInnes et al. (2020) may lead to misinterpretations of the embedding spaces, they can provide visual support to our cross-lingual alignment results. To illustrate this, we fit a UMAP projection on the FLORES devset and plot one randomly sampled English sentence alongside its translations, with the hard negatives from Chen et al. (2023b). To ensure fairness, we only plot the languages common to our baselines. As visualized in Figure 7, ECHO is the only model for which hard negatives are not within the cluster defined around the English sentence.

For a broader perspective, Figure 8 displays 500 sentences from the devset, excluding hard negatives. Across models, clusters consistently form around the same sentence in different languages, with MEXMA, LaBSE, and ECHO exhibiting fewer outliers. However, when hard negatives are introduced (see Figure 7), most models fail to separate them from the target cluster. This visualization highlights the trade-off between xsim and xsim++ performance discussed in section 5: ECHO's contrastive training enables it to push hard negatives away (improving xsim++, as per Figure 7), without compromising its cross-lingual alignment (xsim, as per Figure 8).

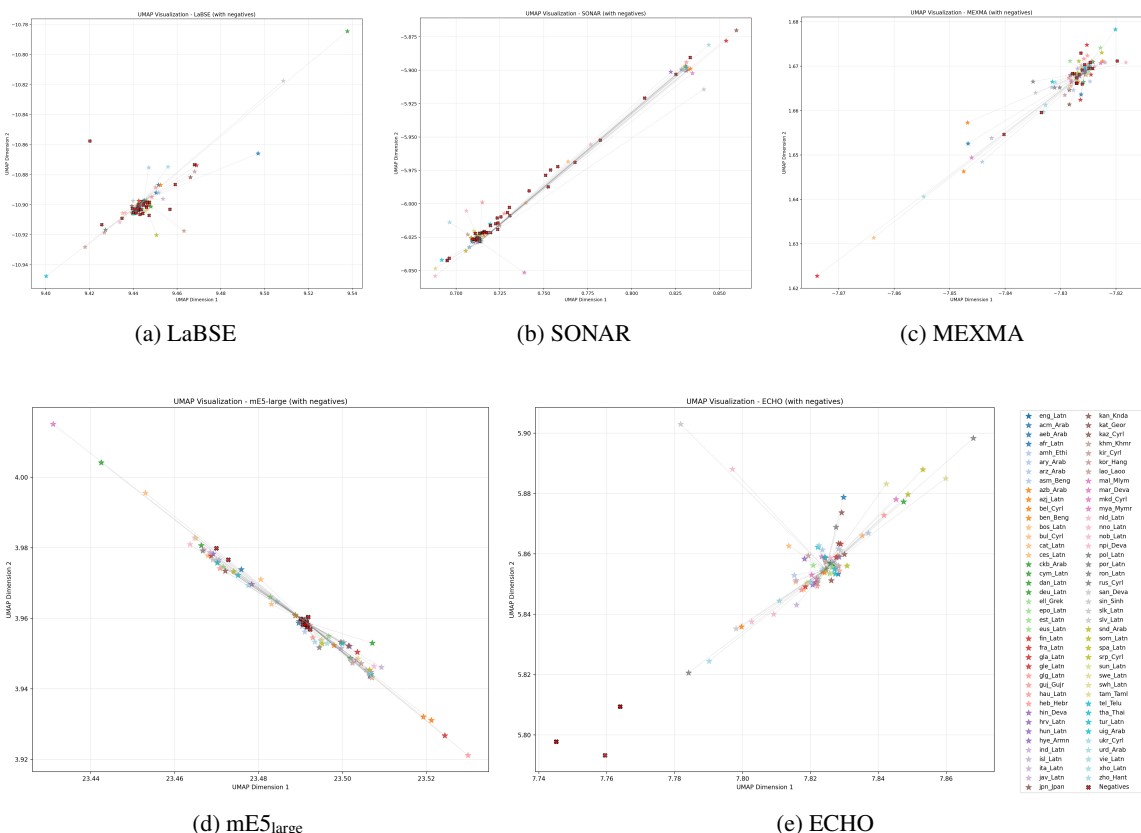

Figure 7: UMAP visualization of the sentence "*During his time with the team, he scored 403 goals in 468 appearances.*" from FLORES devset along closest hard negatives, shown as red crosses. Lines connect the translations to their English counterpart.

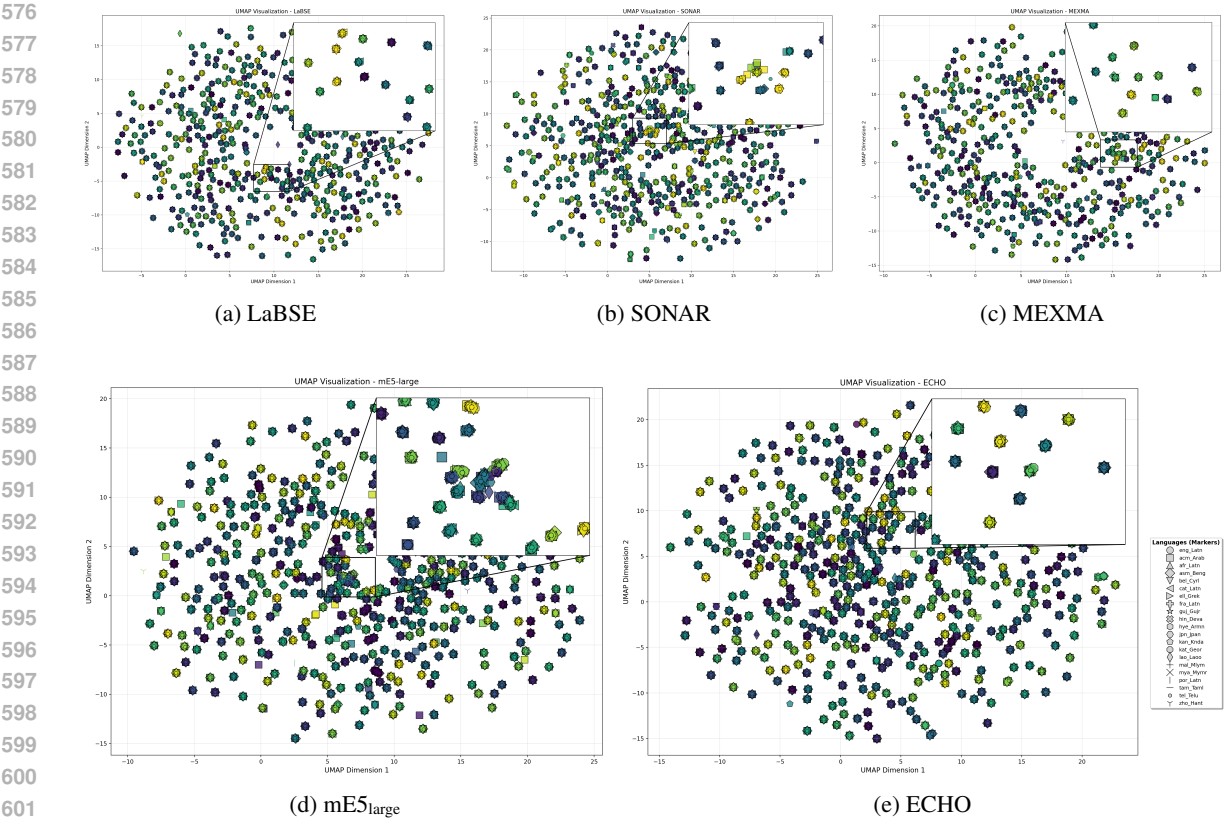

Figure 8: UMAP visualization of the whole space defined by the FLORES devset for 20 languages with different scripts.

