# OpenReview forum: "ECHO: Where Multilingual Sentence Embeddings Speak the Same Language"
_ICLR.cc/2026/Conference — Submitted to ICLR 2026_

### Official Review · Reviewer_Ucgo · 2025-10-29

**Soundness:** 3
**Presentation:** 3
**Contribution:** 2
**Rating:** 4
**Confidence:** 4

**Summary:**

The paper introduces a method for mapping text from any language to a language-agnostic embedding that captures the semantics of the sentence. This is done in 3 training stages:
(1) training a seq2seq translation model
(2) tuning the embeddings obtained from stage 1 by minimizing a contrastive loss plus a translation CE loss
(3) repeating stage 2, but adding hard negative examples to the mix.
The method achieves SoTA results on a variety of evluation settings.

**Strengths:**

- The paper achieves strong results: SoTA in several evals
- Well positioned in prior literature
- Seems useful in practice

**Weaknesses:**

Although I believe the proposed method is practically useful, I'm not convinced the contribution is major enough for ICLR. It's mostly a combination of previous methods, and I didn't see a key innovation beyond that.
I certainly don't want to disparage the work -- I believe there is much value in sharing learnings from method engineering, and the paper achieves very strong results. I applaud the authors for that. But I don't think ICLR is the ideal place to publish this kind of result.

It seems that only a 1B model was trained. Does performance scale with model size? Or has performance saturated already before reaching 1B parameters? In that case, what's the minimum model size that suffices to achieve satisfactory performance?

Some suggestions for improving the paper:
- I found the paper a bit hard to follow in the beginning, as concepts such as "contrastive loss", "hard negatives", "decoder signals" were used without being introduced intuitively. I think the report could be made more appealing by explaining/defining these concepts early on.
- See my questions below. Answering those could help make the manuscript more clear.

**Questions:**

- Line 186: "target sentences incorporate task specification, output language information, and data provenance": was the loss for this part of the target sentence masked? It seems that this would be the right thing to do (or how else should the model know which language it will be asked to decode into?), but I didn't find any statement about this.
- Line 145 says, "We focus on X-to-English directions", but then line 182 says "with more than 5 thousand translation directions". Is this correct? If the 200 languages are considered, wouldn't there be 200 X-to-English directions, rather than 5000?
- Line 255: How many negative examples were used positive example?

---

> ### Author Response · Authors · 2025-11-20
>
> We would like to thank the reviewer for their review, and for highlighting the strong results, usefulness of our paper and its strong position regarding previous literature.
>
> **Response to Weaknesses**
>
> **W1 - Adequacy for ICLR & Novel Contributions:**
> Thank you for your thoughtful feedback regarding the suitability of our work for ICLR. While our approach builds on existing techniques, we introduce several novel components not previously explored in this domain.
>
> Our results demonstrate that combining decoder and contrastive learning objectives marks a significant advance in multilingual representation learning. The decoder loss is critical in shaping the embedding space, yielding a 45% reduction in xsim++ and outperforming all previous encoder-only approaches. To our knowledge, this is the first successful integration of contrastive and decoder objectives, bridging two major paradigms and setting a new direction for future embedding models.
>
> In addition, we introduce a new Split Softmax for hard negatives, a principled approach to false negative removal, and provide comprehensive ablations on objectives, initialization, and data curation. Our model also integrates code and math, extending cross-lingual embeddings to new domains as highlighted by Reviewer 3wu1.
>
> We believe this framework, centered on encoder+decoder+contrastive objectives, should become the foundation for future multilingual embedding research, given its clear empirical advantages over the current encoder+contrastive paradigm. As demonstrated by the spotlight paper NV-Embed at ICLR 2025 (https://openreview.net/forum?id=lgsyLSsDRe), the community values substantial empirical gains achieved through the thoughtful integration of multiple techniques, even when no single algorithmic innovation is present.
>
> Given ICLR’s focus on advances in representation learning, we believe our contributions and results are well-suited for this venue, as also noted by reviewer v5ZA. We hope this clarifies the novelty and relevance of our work, and we appreciate your reconsideration.
>
>
> **W2 - Model Scaling:**
> We focused on a 1B parameter model as it allowed for a consistent comparison with prior work and aligned with the scope and resources of our study. We have not trained smaller scale models due to the non-availability of smaller LLaMA models, with 1B being the smallest.
> However, prior literature (e.g., E5-Mistral) indicates that scaling model size typically leads to further performance improvements. Exploring the impact of model size and alternative architectures is a valuable direction for future research, but outside the scope of this work.
>
>
> **W3 - Presentation Improvements:**
> Thank you for your valuable feedback. We will improve the contextualization of concepts to make it easier for new readers, introducing key terms like "contrastive loss," "hard negatives," and "decoder signals" more intuitively early in the paper.
>
>
> **Response to Questions**
>
> **Question 1 (Line 186 - Loss Masking):** During early experimentation we tried both excluding the prompt from the loss and not removing it. Results were close but we did see a small boost in performance when not excluding it, so we **do include the prompt in the loss.**
> Regarding your comment "or how else should the model know which language it will be asked to decode into?" - the prompt is always present as context when generating, regardless of whether it's masked in the loss. Therefore, the model knows which language to decode into at inference time.
>
> **Question 2 (Lines 145 & 182 - Translation Directions):**
> For the first stage of seq2seq pretraining we consider all x-y directions, but for the remaining stages we do x-eng only. We thank the reviewer for bringing this up and will clarify it in the paper.
>
> **Question 3 (Line 255 - Number of Negatives):**
> 5 negatives per positive sample. It is mentioned in L261.
>
> We thank the reviewer again for highlighting our strong results and submission, and we hope we cleared up how our new representation learning approach fits into ICLR.

---

> > ### Comment · Reviewer_Ucgo · 2025-11-27
> >
> > Thanks for the detailed answer.
> > I am more inclined to see the paper as an appropriate fit for ICLR now.
> > I'd still like to keep my recommendation of 4, because, given the incremental nature of the work (I don't mean this in a negative way), I think the paper could benefit from a more expansive evaluation, including different model sizes and families. This way, practitioners would be able to benefit from the work significantly more.
> > It's solid work, and I believe it can be accepted with some extra effort.

---

> ### Author Response · Authors · 2025-11-27
>
> We thank the reviewer for engaging and raising their concern regarding the model size potentially hindering adoption by practitioners, as well as model families.
>
> We strongly believe that exploring other model families is outside the scope of our work (as for the NV-Embed paper we referenced above, as well as most other state of the art embedding approaches). However, we do agree with the reviewer regarding different scales and exploring performance saturation. Coincidentally we did train smaller model sizes using model pruning, but initially we didn’t consider it a fit for an ICLR paper, hence our previous answer. Upon reflection given your comment, we see the value in smaller models, especially to enable practitioners to benefit from our work at lower compute scales. Therefore we incorporate a section in the paper (Section C.6) explaining how they were trained and reporting on their performance.
> Here is the table that is now added to the paper:
> |     Model    | Size | All Results - xsim | All Results - xsim++ | 80 Languages - xsim | 80 Languages - xsim++ |
> |:------------:|:----:|:------------------:|:--------------------:|:-------------------:|:---------------------:|
> | ECHO (Large) | 1.5B |               0.99 |                 8.12 |                0.07 |                   3.9 |
> |    Medium    | 1.1B |               1.18 |                 8.78 |                0.07 |                  4.21 |
> |     Small    | 806M |               1.23 |                 9.01 |                0.08 |                  4.23 |
> |     Tiny     | 385M |               1.61 |                11.71 |                0.09 |                  5.13 |
>
> An important feature of these models besides just scale is that we ensured their spaces are compatible across sizes. That means one can encode with Echo-Small and use the decoder explained in Section 4.5, or similarly a classifier trained on any of the sizes will, within a certain range, transfer its performance across sizes.
>
> We hope this solves the concern of the reviewer regarding practitioner adoption, and provide a fuller picture about performance saturation, and hope they will raise their score.

---

### Official Review · Reviewer_3wu1 · 2025-10-31

**Soundness:** 3
**Presentation:** 3
**Contribution:** 3
**Rating:** 6
**Confidence:** 3

**Summary:**

This paper introduces ECHO, a novel cross-lingual sentence encoder that repurposes a pretrained Large Language Model  into a bottleneck encoder-decoder architecture. ECHO is trained in three stages:
(1) sequence-to-sequence pre-training on translation tasks,
(2) contrastive fine-tuning with in-batch negatives,
(3) continued contrastive fine-tuning with synthetically generated hard negatives.
The model is evaluated on multilingual alignment, downstream classification and pair classification tasks, and cross-lingual transfer.
ECHO achieves state-of-the-art results in multilingual alignment across 200 languages and demonstrates strong cross-lingual transfer capabilities.

**Strengths:**

1. Give credit to the author to conduct comprehensive evaluation, covering 200 languages and multiple tasks, including multilingual alignment, downstream classification, pair classification, and cross-lingual transfer. The results show consistent improvements over strong baselines (SONAR, MEXMA, LaBSE, mE5large).
2. I like the integration of code and math: The inclusion of code and math data, along with natural language, is innovative. The syntax-aware segmentation of code and the generation of natural language descriptions for code and math expressions add to the modality-agnostic nature of the embeddings.

**Weaknesses:**

1. I am a little confused about hard negative sampling for Code/Math, the method for code and math ("mine the top 5 negatives over a pool of 200k candidates") is less clear and straigthforward. It would be helpful to specify what model is used for this mining and what the nature of these "hard negatives" are.
2. The paper could be benefited by more error analysis. Providing examples where ECHO fails (e.g., low-resource languages, hard negatives, or code/math snippets) would help understand limitations and guide future work.

**Questions:**

see weakness part

---

> ### Author Response · Authors · 2025-11-20
> **1/2**
>
> We thank the reviewer for highlighting the comprehensive evaluation we did, in the number of languages and tasks, comparing to strong baselines and also appreciating the inclusion of code and math which is innovative.
>
> **Response to Weaknesses/Questions**
>
> **Q1. Hard Negative Sampling for Code/Math:**
> The previous sentence (L846) to the one you mentioned states: "mine hard negatives using the ECHO checkpoint trained in Section 4.3 before hard negatives are introduced". We see how the order in which things are stated may be confusing and we will make sure to rephrase this section to better explain hard negative mining. Regarding their nature, given a pool of 200k negatives, we sample the closest 5 "explanations" to each code snippet. **An example from our corpus:**
> ```
> Python: temp = pd.read_csv(item, header = None, dtype = float)
> English: The pandas library is used to read a csv file specified by the item variable into a temporary variable named temp with the header set to None and the data type set to float.
> Hard Negatives:
>     * The code reads data into a variable named df using the pandas function read_csv.
>     * The variable temp_mean is assigned the result of pandas' read_csv function applied to the string conversion of config's attribute.
>     * The pandas library is used to read a csv file named "data.csv" located in the "/data" directory into a variable named df using the read_csv function.
>     * The pandas library, referred to as pd, reads a comma-separated values file named 'data/time_series_19-covid-Confirmed.csv' into a variable named df using the read_csv function.
>     * The pandas library, referred to as pd, reads a comma-separated values file named 'tmdb-movies.csv' into a dataframe variable named df using the read_csv function.
> ```
> We have included these examples in the paper, and thank the reviewer for their suggestion on how to improve clarity in this section.
>
> **Q2. Error Analysis:**
>  Following the reviewer concern, we performed some qualitative analysis of the errors of ECHO. By inspection, ECHO's mistakes look to be related to unit conversion, selecting the sentence where the actual numerical value matches, i.e. matching "15 cm" to "15 inches" instead of "6 inches". We theorize this is due to our hard negatives focused on matching the values, but lead to errors when the translation transforms the units. Meanwhile we see SONAR and MEXMA make mistakes related to both values and semantics (may/will, white/black), such as the examples provided below. We would also like to highlight that Tables 16 and 17 have a full breakdown of results per language of xsim and xsim++, to understand the performance in both high and low resource languages.
>
> ECHO:
> ```
> {
>   "source_sentence": "O Corpo de Engenheiros dos EUA estimou que 15 cm de chuva podem romper os diques anteriormente danificados.",
>   "desired_retrieved_sentence": "The U.S. Corps of Engineers estimated that 6 inches of rainfall could breach the previously damaged levees.",
>   "actual_retrieved_sentence": "The U.S. Corps of Engineers estimated that 15 inches of rainfall could breach the previously damaged levees.",
>   "score_desired": 0.8148834705352783,
>   "score_actual": 0.859208345413208
> }
> {
>   "source_sentence": "Os limites de velocidade anunciados sao visivelmente mais baixos do que nas secoes anteriores e subsequentes - comumente 55-65 km/h - e a estrita obediencia a eles e ainda mais importante do que o contrario.",
>   "desired_retrieved_sentence": "Posted speed limits are noticeably lower than in previous and subsequent sections  commonly 35-40 mph (56-64 km/h)  and strict obedience to them is even more important than otherwise.",
>   "actual_retrieved_sentence": "Posted speed limits are noticeably lower than in previous and subsequent sections  commonly 35-90 mph (56-64 km/h)  and strict obedience to them is even more important than otherwise.",
>   "score_desired": 0.8505960702896118,
>   "score_actual": 0.8566824197769165
> }
> ```
>
> MEXMA:
> ```
> {
>   "source_sentence": "O Corpo de Engenheiros dos EUA estimou que 15 cm de chuva podem romper os diques anteriormente danificados.",
>   "desired_retrieved_sentence": "The U.S. Corps of Engineers estimated that 6 inches of rainfall could breach the previously damaged levees.",
>   "actual_retrieved_sentence": "The U.S. Corps of Engineers estimated that 15 inches of rainfall could breach the previously damaged levees.",
>   "score_desired": 0.8599843382835388,
>   "score_actual": 0.8785548210144043
> }
> {
>   "source_sentence": "Reportagens televisivas divulgam a fumaca esbranquicada saindo da planta.",
>   "desired_retrieved_sentence": "Television reports show white smoke coming from the plant.",
>   "actual_retrieved_sentence": "Television reports show black smoke coming from the plant.",
>   "score_desired": 0.7506682872772217,
>   "score_actual": 0.7513110637664795
> }
> ```

---

> > ### Author Response · Authors · 2025-11-20
> > **2/2**
> >
> > SONAR:
> > ```
> > {
> >   "source_sentence": "No periodo de um ano, uma pessoa infectada pode infectar entre 10 e 15 contatos proximos.",
> >   "desired_retrieved_sentence": "In one year's time, an infected person may infect 10 to 15 close contacts.",
> >   "actual_retrieved_sentence": "In one year's time, an infected person will infect 10 to 15 close contacts.",
> >   "score_desired": 0.900607168674469,
> >   "score_actual": 0.9051467776298523
> > }
> > {
> >   "source_sentence": "Aconteceu novamente no mesmo mes em Mashhad, outro aviao comercial entrou em uma pista e atingiu uma parede, matando dezessete pessoas.",
> >   "desired_retrieved_sentence": "The same month saw another airliner overrun a runway at Mashhad and strike a wall, killing seventeen.",
> >   "actual_retrieved_sentence": "The same month did not saw another airliner overrun a runway at Mashhad and strike a wall, killing seventeen.",
> >   "score_desired": 0.6934364438056946,
> >   "score_actual": 0.7041366696357727
> > }
> > ```
> >
> > As with the hard negative examples, we will include these in the Appendix. We agree with the reviewer that these strengthen and contextualize our contribution, and we hope to have addressed the reviewer concerns.
> >
> > As a recap, we provided additional details and examples to the reviewer to clarify their questions, and ask whether they have any additional comments or questions. Otherwise, we ask the reviewer to consider raising their score.

---

### Official Review · Reviewer_v5ZA · 2025-11-01

**Soundness:** 2
**Presentation:** 1
**Contribution:** 2
**Rating:** 2
**Confidence:** 4

**Summary:**

This paper proposes ECHO, a three-stage framework that finetune a encoder-decoder model intitilaized with LLMs for multilingual sentence embeddings. Using parallel corpus, the proposed ECHO finetune the model with translation objective (stage 1), contrastive objective in the encoder output representation space (stage 2) and contrastive learning with hard negative (stage 3). Experiments on cross-lingual retrieval and classfication show emprical improvements compared to the baseline.

**Strengths:**

1. Multilingual sentence embedding is an important tasks that have many applications. It aligns with the scope of ICLR too.
2. Emprical results show improvements compared to previous embedders.
3. Non-natural languages including math and code are included in the experiments. This is useful to study multilingual embedding in a broader scope.

**Weaknesses:**

1. The English writing needs improvement. A revising by native speaker or LLMs can be helpful.
2. The organization of this paper is not clear. For example, the motivation and justification for using encoder-decoder framework is not clearly stated. For example, "...they largely overlook cross-lingual transfer..." is not a informative argument to introduce the limitation of previous work, especially the proposed method has significant overlap with baselines such as SONAR [1]. A more structual story-tell can improve this.
3. A main concern is that the novelty of this work seems to be limited. The objectives of machine translation along with contrastive learning has been studied in previous works such as [1] and [2] and etc. Sorely utilzing LLMs as backbone lacks of technical contribution. Hard negative for contrastive learning is also insufficient to be a contribution.
4. In addition to limitation 3, the baselines and proposed ECHO are built on different backbones, and LLaMA3 is more powerful initillay hench the comparision is not convincing enough. More details on the training settings including training data use to show that the baselines and ECHO are comparable will be helpful.
5. As for the experiments settings, only LLaMA3 is used so the emprical results are not model-agnositic which limits the contribution. In addition, LLaMA3 is a decoder LLM and why utilizing it for both encoder and decoder framework is not justified.
6. The ablation study is weak. The three stage training is the core design of ECHO, however, training with only state 2 or stage 3 are not evaluated to show how those stages affect the behavior of the framework. Table 9 in appedix seems not introduce the detailed experimental setting and also missing settings metioned above but just incremental experiments from stage 1 to 1+2 and 1+2+3.


Minor comments:
1. line 30: citations are broken
2. What is the implication of "speak the same language" in the title?

[1] SONAR: sentence-level multimodal and language-agnostic representations.
[2] Enhancing Cross-lingual Sentence Embedding for Low-resource Languages with Word Alignment, NAACL 2024 findings.

**Questions:**

1. The decoder is designed for probing the sentence embedding, which is useful. However, the focus of this paper is multilingual embedding and the decoder design is not nessasary, for example, we could just fine-tune a encoder-only embedder and train a seperate probing decoder for this purpose. I would appreciate more explanation on this issue.

2. What is the essential difference between ECHO and SONAR? I beleive the answer to this question will highlight the contribution of this work better.

---

> ### Author Response · Authors · 2025-11-20
> **1/2**
>
> We thank the reviewer for their comments, and for highlighting that our approach aligns with the scope of ICLR.
>
> We kindly pushback against some of the weaknesses that we believe come from misunderstandings:
>
> **W3 Reconsideration**: The reviewer implies that the use of a decoder trained on MT alongside contrastive learning to learn cross-lingual sentence embeddings "has been studied in previous works such as [1] and [2] and etc." However neither of the examples provided adhere to that argument:
> * SONAR [1] doesn't use any contrastive signal.
> * WACSE [2] is just an encoder, not trained with generative Machine Translation.
>
> Therefore we reiterate the innovative aspect of ECHO, and suggest a reconsideration, especially since this is "A main concern" for the reviewer. We have cited WACSE [2] to better contextualize our work, and highlight our differences.
>
> **W4 Additional Evidence**: Table 9(b) precisely tackles the reviewer's concern. We follow the **SONAR recipe** and train solely using MSE as a signal, which showcases a **~4 point difference on xsim++**. Nevertheless to provide a more recent baseline, we also now trained a model with the **MEXMA approach** using the same backbone from step 1 (Seq2Seq). We obtained an xsim++ of 12, still **~3 points higher than ours**. Regarding training data for other models, L120/L121 highlight that previous models also used NLLB data to train their models, as we also do.
>
> We have better highlighted these results in the paper, and hope this is enough to ease the concerns of the reviewer.
>
> **W6 Clarification**: The reviewer may have missed Table 11 (b) which provides exactly what they suggest, skipping step 1, and training directly on step 2. This has a ~3% xsim++ drop, further motivating our approach. We will make sure to highlight them more prominently on the main body.
>
> Regarding the absence of "Stage 3 only" evaluation, hard negatives as a later stage follows established practice and does not require separate ablation. This two-phase approach (base contrastive training → hard negative refinement) is well-established:
> * GTE (Li et al., 2023): Applies hard negative mining in subsequent training phases
> * mE5 (Wang et al., 2024b): Uses synthetic hard negatives in a later finetuning stage after initial contrastive training
> * GISTEmbed (Solatorio, 2024): Introduces guided in-sample hard negative selection as a refinement step
> * E5-mistral (Wang et al., 2024): Follows multi-stage training where hard negatives are introduced after establishing base representations
>
> The experimental setup is the same across experiments unless specified otherwise, as defined in Section 4.6, including those in the Appendix. As with most previous points, we believe the paper generally addresses the reviewer's concern and ask to reconsider this as a Weakness.
>
>
> Other comments:
>
> **W5**: There is extensive research using LLMs as encoders, as we highlight in L65-L68 by citing LLM2Vec, and as both encoder and decoder, such as Encoder-Decoder Gemma cited there as well, which justify their use in ECHO. We selected LLaMA3 as a standard open source LLM. While we agree that it would be a nice addition to try other LLMs as backbone, it felt outside the scope of our work.
>
> **W1 & W2**: We thank the reviewer for the feedback, we will leverage the extra page of the camera ready to expand the story-telling and revise the English writing.
>
> Response to Questions:
>
> **Question 1**: The question stems from missing a key aspect of our approach. The cross-entropy loss from the decoder is a major training signal as clearly emphasized throughout the paper (L220-L221, L236, L394-395, Table 9 (a), Section C.1…). As stated in the paper: “Adding the Decoder loss to the contrastive learning stage in subsection 4.3 reduces xsim++ error by 45%”.
>
> **Question 2**: SONAR and ECHO are similar in the use of a Decoder signal during training, however it differs from it in virtually all other aspects. While SONAR demonstrated the value of using a decoder to learn sentence representations, it suffered from embedding collapse due to its MSE alignment objective. ECHO builds on SONAR’s encoder-decoder approach but addresses these shortcomings by employing contrastive learning, which significantly improves representation quality and mitigates collapse. We additionally show that the decoder training signal improves upon simple contrastive learning. This sets what should be the future paradigm of embedding representation learning, coupling the contrastive loss with a decoder training signal. Additionally, ECHO introduces several novel aspects compared to SONAR: it leverages an LLM-based architecture, supports code and math, enables prompting, uses a more improved tokenizer covering 200 languages, among others.

---

> > ### Author Response · Authors · 2025-11-20
> > **2/2**
> >
> > To recap, we appreciate the reviewer highlighting that our approach aligns with the scope of ICLR, and reiterate that, as mentioned above, most of the concerns expressed by the reviewer were already addressed or justified in the paper itself. We also provided some additional experiments to further illustrate our point. We reiterate our commitment to improve the paper's writing and clarity to avoid any misunderstandings, but **hope the reviewer reconsiders their weaknesses and the reflected low score**.

---

### Author Response · Authors · 2025-11-20

We appreciate the reviewers' time and effort in providing feedback on our submission. We have responded to all comments and uploaded a revised manuscript (with changes in blue).

---

### Meta-Review · Area_Chair_7Sfy · 2025-12-30

**Summary:**

A 3-stage model is proposed for multilingual sentence embedding. All reviewers find the proposed method to be practically effective, but novelty is limited as each individual stage has appeared.

**Reviewer Concerns:**

The main concern is novelty, which may be subjective to different reviewers, but the consensus seems to be that this aspect is lacking a little bit.

**Reviewer Scores:**

None is likely to change.

---

### Decision · Program_Chairs · 2026-01-26

Reject